# A lysozyme with altered substrate specificity facilitates prey cell exit by the periplasmic predator *Bdellovibrio bacteriovorus*

Christopher J. Harding [1,3], Simona G. Huwiler [2,3], Hannah Somers[2], Carey Lambert[2], Luke J. Ray[2], Rob Till[2], Georgina Taylor[2], Patrick J. Moynihan [1], R. Elizabeth Sockett [2✉] & Andrew L. Lovering [1✉]

Lysozymes are among the best-characterized enzymes, acting upon the cell wall substrate peptidoglycan. Here, examining the invasive bacterial periplasmic predator *Bdellovibrio bacteriovorus*, we report a diversified lysozyme, DslA, which acts, unusually, upon (GlcNAc-) deacetylated peptidoglycan. *B. bacteriovorus* are known to deacetylate the peptidoglycan of the prey bacterium, generating an important chemical difference between prey and self walls and implying usage of a putative deacetyl-specific "exit enzyme". DslA performs this role, and ΔDslA strains exhibit a delay in leaving from prey. The structure of DslA reveals a modified lysozyme superfamily fold, with several adaptations. Biochemical assays confirm DslA specificity for deacetylated cell wall, and usage of two glutamate residues for catalysis. Exogenous DslA, added ex vivo, is able to prematurely liberate *B. bacteriovorus* from prey, part-way through the predatory lifecycle. We define a mechanism for specificity that invokes steric selection, and use the resultant motif to identify wider DslA homologues.

[1] Institute for Microbiology and Infection, School of Biosciences, University of Birmingham, Birmingham B15 2TT, UK. [2] Medical School, School of Life Sciences, Queen's Medical Centre, University of Nottingham, Nottingham NG7 2UH, UK. [3] These authors contributed equally: Christopher J. Harding, Simona G. Huwiler. ✉email: liz.sockett@nottingham.ac.uk; a.lovering@bham.ac.uk

The relationship between lysozyme(s) and the bacterial cell wall is historically prominent in microbiology and biochemistry, starting with the discovery of lysozyme antibacterial activity by Fleming in 1922 (ref. [1]) and progressing to the first elucidation of an enzyme structure by Phillips in 1965 (ref. [2]). From these beginnings, lysozymes are now among the most highly characterized enzymes in biochemistry. Broadly, the susceptibility of the bacterial cell-wall peptidoglycan to lysozyme activity encompasses the usage of lysozyme in mucosal immunity[3], pathogen modification of the wall to evade killing[4], deployment of lysozymes during interbacterial competition[5], and usage of lysozyme/lytic transglycosylase activity in tailoring the cell wall during growth, division, signaling, autolysis, sporulation, and insertion of wall-spanning protein complexes[6]. Lysozymes (together with related lytic transglycosidases, bracketed as muramidases) cleave peptidoglycan strands between the C1 of N-acetylmuramic acid (MurNAc) and C4 of N-acetylglucosamine (GlcNAc) sugars; although modifications of the cell-wall peptide are relatively common, the sugar backbone is much less variable. Modification of peptidoglycan to endow resistance to human lysozyme is common in Gram-positive pathogens which have environmentally exposed cell walls (e.g., *Listeria*, *Streptococcus*, *Staphylococcus* spp.) and takes the form of either O6-acetylation (MurNAc) or N-deacetylation (MurNAc and/or GlcNAc)[7]; both reactions take place on the mature peptidoglycan (PG) polymer. To date, lysozymes with low specificity are known (e.g., cellosyl will turnover both acetylated and deacetylated peptidoglycan)[6] but no enzyme has been characterized that is specific for the N-deacetylated form only.

The intraperiplasmic bacterial predator *Bdellovibrio bacteriovorus* is adept at peptidoglycan manipulation, entering through the outer membrane and wall of other Gram-negative bacterial cells to the inner periplasmic compartment and systematically metabolizing bacterial prey cell components from the inside[8]. Other external bacterial predators of bacteria also encode wall-metabolizing enzymes (*Myxococcus* and *Streptomyces* among the best characterized), but aside from the endpoint of prey cell destruction, *Bdellovibrio* differs in that it has entry, residence and exit lifecycle stages that require more nuanced approaches to define and differentially manipulate chemically related self- and bacterial prey cell structures at different timepoints. Exit from prey requires destruction of several complex cell wall and outer membrane structures by predators, without damaging self.

A series of early papers by Thomashow and Rittenberg established that *B. bacteriovorus* has glycanase and peptidase activities for entry[9], deacetylase(s) that modify the host cell-wall material[10], and activities that solubilize/reattach diaminopimelate and acylate the peptidoglycan with a long-chain fatty acid[11]. More recent investigations have characterized some of these enzymes, finding that entry-secreted D,D-endopeptidases from *B. bacteriovorus* break 3,4 peptide crosslinks in prey PG walls, thereby rounding/softening prey PG and preventing wasteful entry by a second predator[12]. *Bdellovibrio* protects itself during these modifications by producing a protein that complexes and inhibits the peptidases — failure to do so results in suicide[13]. Other wall modifications during predation have been observed via the use of fluorescent D-amino acids, revealing a porthole-like structure that forms during an invasion, and other L,D-transpeptidase-dependent activities that reinforce the prey cell "bdelloplast" while occupied by the *B. bacteriovorus* for its intrabacterial lifecycle[14].

Initial hypotheses that *Bdellovibrio*-catalyzed deacetylation of prey peptidoglycan protected against host autolysins were disproven; instead, deacetylation marks prey (as distinct from the predator, which retains acetylated peptidoglycan) for selective destruction at the end of the intraperiplasmic lifecycle[15]. Mutant predator strains lacking secreted peptidoglycan GlcNAc deacetylases (ΔBd0468 and ΔBd3279) complete predation but leave behind a "ghost" cell of undigested prey cell-wall material[15]. Peptidoglycan isolated from *E. coli* bdelloplasts is N-deacetylated (~60–70% of MurNAc and GlcNAc) and thereby resistant to digestion by conventional lysozymes[10]. Hence, *B. bacteriovorus* and related predators have the potential to encode a deacetylation-specific lysozyme.

Here, we discover a family of *Bdellovibrio* proteins that define a new subgrouping of the lysozyme superfamily and demonstrate that they possess outright specificity for deacetylated peptidoglycan. We solve the structure of family member Bd0314 (that we named DslA), revealing adaptations to the lysozyme fold that sterically restrict the active-site cleft into only accepting polymers deacetylated at the GlcNAc positions. We determine a role for DslA in *B. bacteriovorus* exit from the prey cell and demonstrate that exogenously added enzyme is able to prematurely liberate predators from prey bdelloplasts before completion of the predatory lifecycle.

## Results

**Lysozyme-like gene *bd0314* is expressed during prey exit phase**. In the genome of *B. bacteriovorus*, four divergent/cryptic lysozyme homologs are encoded, Bd0314 (DslA), Bd1413 (DslB), Bd1440 (DslC), and a more distal member Bd1411 (22% identity to DslA, in comparison to DslB and DslC which are 30% identical to DslA and 60% identical to each other). Transcriptional analysis revealed that *bd1411* was upregulated in the attachment phase (about 15 min after infection), whereas *bd1413* and *bd1440* transcription was not differentially regulated at specific timepoints of the predatory lifecycle of *B. bacteriovorus*, but expressed throughout (Supplementary Fig. 1). In contrast, *bd0314* was upregulated in the late stages (180–240 min after infection) of the predatory lifecycle — around prey exit (Fig. 1). Therefore, Bd0314 (DlsA) was hypothesized to be involved in the exit and release of new progeny *B. bacteriovorus* cells from the inner periplasm of prey bdelloplasts. Further analysis indicated that *bd0314* was transcribed alone and not with flanking genes (Supplementary Fig. 2).

The roles of each of the different lysozymes in the predatory lifecycle were addressed by gene deletion (Δ*bd0314*, Δ*bd1411*, Δ*bd1440*) or inactivation of an active-site residue (E151Q in Bd1413, resulting in *bd1413EQmc*). The deletion of *bd0314* resulted in a significant delay in prey exit time and is studied further as described below. In comparison, time-lapse microscopy of the entry phase revealed a significantly increased entry time for a mutant deleted for the early-expressed (more distantly related) *bd1411*, while Δ*bd0314* also indicated a small potential increase (*p*-value 0.0388) in entry time (Supplementary Fig. 3). However, the inactivation of the active-site in Bd1413 or deletion of Bd1440 did not delay the entry process. These data indicate a primary role for Bd1411 in prey entry (which is not studied further in this paper) and an important role for Bd0314 in prey exit.

**Bd0314 is important in prey exit**. Epifluorescence time-lapse microscopy revealed a delay in the overall exit time of *B. bacteriovorus* Δ*bd0314* in comparison to the wild-type HD100 strain (Fig. 2). The median exit time of the wild type was 65 min, whereas it took Δ*bd0314* a median of 100 min from predator progeny septation in the bdelloplast until the first progeny cell escaped the prey. Complementation of Δ*bd0314* by the single crossover of pK18-*bd0314* with the wild type *bd0314* significantly reduced exit time relative to the mutant strain. Complementation did not completely restore the exit phenotype back to wild-type level. This is likely a result of the presence of two *bd0314* promoter regions (one in front of the wild-type *bd0314* and one in

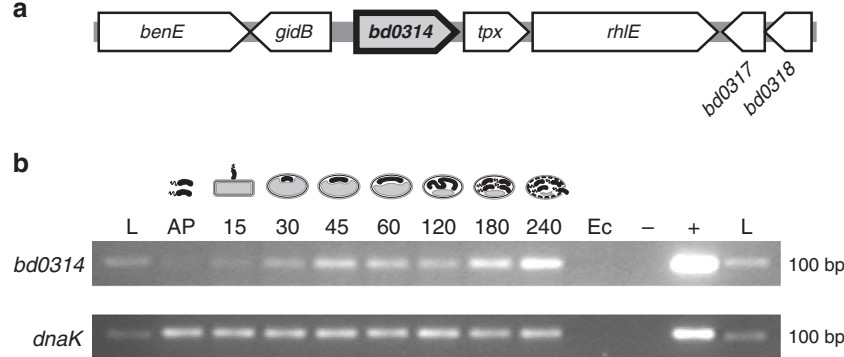

**Fig. 1 Transcriptional expression and genomic context of lysozyme homolog-encoding bd0314. a** Scheme of the genomic context of bd0314 including surrounding genes predicted to encode a benzoate membrane transport protein (benE), a glucose-inhibited division protein B (gidB), a thiol peroxidase (tpx), an ATP-dependent RNA helicase (rhlE), and two hypothetical proteins (bd0317, bd0318). **b** Transcriptional analysis by reverse transcriptase PCR showing the expression of lysozyme gene homolog bd0314 and housekeeping gene dnaK, during the host-dependent lifecycle of B. bacteriovorus HD100 as depicted in drawings above the agarose gel. AP: attack phase HD100 cells, 15–240: timepoint in minutes after the invasion, Ec: E. coli S17-1 prey cells, −: negative control with $H_2O$, +: positive control with genomic DNA of B. bacteriovorous HD100, L: 100 bp DNA ladder. This figure shows one of two biological repeats with similar results.

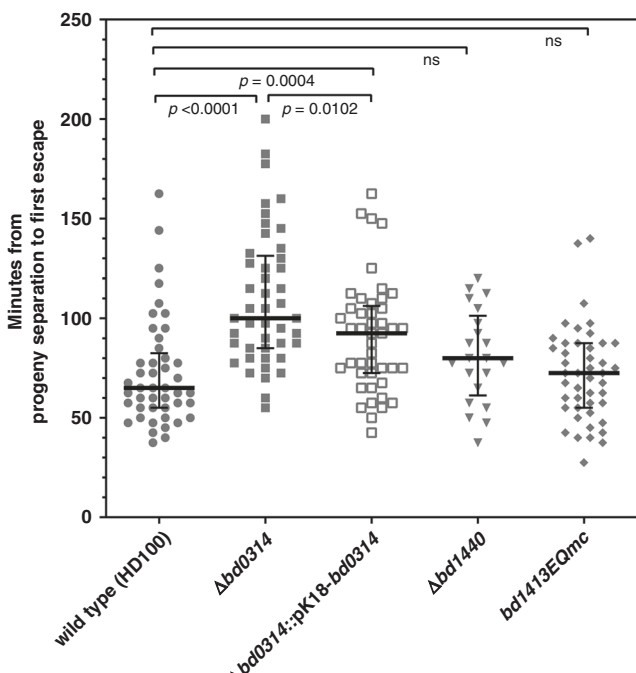

**Fig. 2 Δbd0314 shows a delayed prey exit phenotype.** Comparison of time required from progeny separation to the first exit of B. bacteriovorus HD100 wild type vs. Dsl- family mutant strains (Δbd0314, Δbd1440, and bd1413EQmc) and complemented strain (Δbd0314::pK18-bd0314) from prey cell bdelloplast remnants monitored by epifluorescence time-lapse microscopy. Bdelloplasts were placed on a 1% agarose slide ~3 h post-infection of E. coli S17-1::pMAL-p2_mCherry (generating fluorescent prey periplasm and cytoplasm) and observed for another 4 h 40 min to determine backlit predator progeny septation and any exit[34]. Median values and interquartile range are shown as black lines. Data originate from at least two independent biological replicates (n = 45 for each strain, except for Δbd1440, where n = 21). The two-tailed p-values are derived from the Mann–Whitney test (ns non-significant, p-values are presented in the figure).

front of the deletion remnant of bd0314), in the complementation strain, which may have reduced levels of bd0314 gene expression relative to wild-type HD100 strain. Predatory strains with a mutation/deletion of bd1413 (dslB) or bd1440 (dslC) did not show

a statistically significant increase in exit time compared to wild type. As median exit times of these strains are between the ones of wild type and Δbd0314 (80 min for Δbd1440 and 72.5 min for bd1413EQmc), a minor role for DslB and DslC in prey exit may be implied. The progeny cells of Δbd0314 are trapped for longer within the bdelloplast at the end of predation, which confirms a major role for Bd0314 in the exit process. This implies that the DslA enzyme may act upon the prey peptidoglycan, wherein the GlcNAc residues had been previously deacetylated at the prey invasion stage by B. bacteriovorus Bd3279 and Bd0468 enzymes[15]. For double GlcNAc-deacetylase B. bacteriovorus mutant Δbd0468 Δbd3279 prey cell-sized "ghost" remnants were abundant in cultures after predation[15]. These large ghost remnants were not seen for the wild type and Δbd0314 mutant strain, testing with wheat-germ-agglutinin-Alexa-Fluor-350 conjugate staining (Supplementary Figs. 4 and 5).

*DslA structure determination*: Exit-specific lysozyme Bd0314 was named DslA for deacetylation-specific lysozyme based on its specific activity as shown later in this article and to distinguish it from the 'original' lysozymes unable to act on deacetylated peptidoglycan[6]. We were able to successfully purify the enzymatic domain of DslA by omitting the lipobox and ~35 aa region of low complexity/disorder, fusing region R74-K254 to maltose-binding protein (later cleaved). This construct yielded two independent crystal forms, diffracting to 1.26 and 1.36 Å resolution (Table 1). Attempts to solve these via molecular replacement with "conventional" lysozyme structures failed, indicating potential divergence from the superfamily fold.

Large crystals of a catalytic E154Q mutant enabled us to solve the structure via Sulfur-SAD phasing, finding two disulfides and two methionine residues, tracing residues 74–254 inclusive (Fig. 3). The fold is comprised of seven major α-helices (A-G), several smaller $3_{10}$ helices, an elongated β-hairpin (b1–2) and a smaller β-hairpin (b3-4). The fold is stabilized by disulfide bonds between C117:C253 and C182:C199. The secondary structure elements create a classical two-lobed architecture, with helix E and β1–2 sat below a pronounced active-site cleft (measuring ~25 Å across, ~15 Å deep). The lysozyme superfamily has a conserved ES motif at the C terminus of a central α-helix, which is readily identifiable as E143/S144 on DslA helix D (Fig. 3). A second acidic residue, E154 projects upward from the β-sheet floor of the cleft, and its conformation is stabilized by a hydrogen bond to S166 and packing effects mediated by the hydrophobic sidechains of Y152 and F156. The pose of E154 is further influenced by

**Table. 1 Data collection and refinement statistics.**

|  | DslA form 1 | DslA form 2 | DslA E143Q | DslA E154Q |
|---|---|---|---|---|
| Accession code | 6TA9 | 6TAB | 6TAD | 6TAF |
| Data collection |  |  |  |  |
| Space group | $P2_12_12$ | $P2_12_12_1$ | $P2_12_12_1$ | $P2_12_12$ |
| Cell dimensions | 84.7, 47.8, 49.2 | 48.9, 64.7, 82.7 | 48.1, 62.1, 80.9 | 84.7, 47.9, 49.2 |
| $a, b, c$ (Å) | 90, 90, 90 | 90, 90, 90 | 90, 90, 90 | 90, 90, 90 |
| $\alpha, \beta, \gamma$ (°) |  |  |  |  |
| Resolution (Å) | 1.36 | 1.26 | 1.82 | 1.37 |
| $R_{sym}$ | 0.088 (0.785)[a] | 0.068 (1.786) | 0.161 (0.592) | 0.137 (1.713) |
| $R_{pim}$ | 0.026 (0.279) | 0.020 (0.696) | 0.022 (0.227) | 0.040 (0.489) |
| $I/\sigma I$ | 16 (2) | 13 (1.2) | 20.1 (2.7) | 12.2 (2.2) |
| $CC_{1/2}$ | 0.998 (0.804) | 0.999 (0.509) | 0.998 (0.722) | 0.994 (0.617) |
| Completeness (%) | 98.1 (82.8) | 99.8 (98.4) | 87.65 (21.1) | 98.4 (100) |
| Redundancy | 12.4 (8.5) | 11.4 (7.5) | 49.5 (8.6) | 12.5 (13) |
| Refinement |  |  |  |  |
| $R_{work}/R_{free}$ | 14.0/18.0 | 15.0/17.7 | 17.2/21.0 | 17.7/19.5 |
| R.m.s. deviations |  |  |  |  |
| Bond lengths (Å) | 0.006 | 0.014 | 0.006 | 0.005 |
| Bond angles (°) | 1.03 | 1.55 | 0.79 | 1.04 |

[a]Values in parentheses are for highest-resolution shell. All data sets derived from a single crystal each.

interruption of main chain hydrogen-bonding: the backbone carbonyl groups of E154 and D155 point downward, not participating in β-hairpin formation. The net effect creates a bulge (Fig. 3c) and allows the sidechain of E154 to face upward, directly across from E143.

Comparison of the DslA fold with structural homologs using DALI[16] confirmed a partial, but importantly not total, similarity to lysozyme superfamily members. Secondary structure matching confirms that the DslA helix-rich upper lobe is more similar to lytic transglycosylases, e.g., MltF (PDB 4OXV, Z-score 9.8, rmsd 3.5 Å, 122 aa alignment at 15% identity; unpublished but similar to this study)[17] but this is not significantly higher than a generalized agreement to all lysozymes (e.g., c-type human lysozyme[18], 1LZS, 8.3, 2.6 Å, 104 aa, 13%). The comparison of all lysozyme superfamily members, including chitinases and chitosanases, noted the conservation of helices D and F, the ES motif and the β-floor of the cleft[19]; these features are the only factors in agreement when superimposing DslA with the more distantly related GH24 and GH46 enzymes (Fig. 3d). This broader analysis indicates that DslA has four unique regions not shared with any lysozyme subfamily: 73–87 (N terminus, helix A), 99–124 (loop region and helix C), 154–164 (the extended β1–2 hairpin past E154), and 214–224 (the β3-4 loop), the latter two features extending the depth of the active-site cleft. The tip of the extended β1–2 hairpin is stabilized both internally (by the sidechain of D158) and externally (by the sidechain of E175 from the N terminus of helix E).

Classical lysozyme superfamily catalysis utilizes two acidic residues for hydrolysis or one residue for the GH23/GH102/GH103/GH104 lytic transglycosylases that generate cyclic 1,6-anhydro-MurNAc. The first, as part of the ES motif, is invariable (Fig. 3), but the position of the second alters dependent on retaining versus inverting mechanisms[20]. The second acidic residue of human lysozyme, D53, lies on β2 and is equivalent to DslA S166 (Fig. 3e). In contrast, the DslA active-site residue E154 is situated on β1 and, despite a relative twisting of the hairpin, is spatially equivalent to the second catalytic residue of v-type lysozymes (e.g., 2LZM D20) and chitosanases (e.g., 4OLT D43) as shown in the zoomed-in comparison of Fig. 3e. In both these other enzyme classes, the second acid is coordinated by a Thr residue, whose equivalent in DslA is S166. The average distance between the DslA E143 and E154 sidechain carboxylates is 8 Å, in

agreement with the inverting superfamily members that place the second catalytic acid on β1 (ref. [19]). All four of our DslA structures (two wild-type, E143Q, E154Q) are highly similar, other than a relatively small flexation of the β1–2 hairpin in the E154Q variant (Supplementary Fig. 6). The key differences between our structure and conventional lysozymes explain (see the Discussion section) why we went on to find that they have specificity for PG containing deacetylated GlcNAc, unlike conventional lysozymes which are specific for PG containing acetylated GlcNAc.

*An enzymatic assay confirms GlcNAc-deacetylase specificity of DslA*: We next sought to confirm DslA specificity for GlcNAc-deacetylated peptidoglycan, through use of an enzymatic assay (Fig. 4). This assay uses confirmed *B. bacteriovorus* GlcNAc-deacetylase proteins[15] Bd0468 and Bd3279 to generate a modified PG substrate from originally acetylated *E. coli* peptidoglycan labeled with a FITC fluorophore. Hydrolysis was monitored via an increase in fluorescence originating from the conversion of insoluble peptidoglycan to soluble smaller fragments. The assay confirmed the known specificity of hen egg-white lysozyme (HEWL) for acetylated substrate, and lack of specificity for the relatively open active site of mutanolysin[6]. Partial activity for HEWL against the GlcNAc-deacetylase-treated material was indicative of our in vitro prior deacetylation of *E. coli* PG not running to completion, and illustrative of the difficulties of reconstituting *B. bacteriovorus* activities in vitro outside of the bdelloplast. Despite this, DslA was clearly inactive against native, acetylated peptidoglycan, but displayed strong activity against the deacetylated material; this level of activity was on a similar scale to that of HEWL and mutanolysin with their known substrates.

Using the DslA structure as a guide, we next monitored the activity of DslA mutants for the putative active-site residues. Variants E143Q and E154Q clearly fold and retain a similar active-site architecture (as judged by our high-resolution structures herein, Supplementary Fig. 6), but display no notable activity (Fig. 4, middle), confirming their participation in catalysis. Variant Y228A has residual activity on GlcNAc-deacetylated peptidoglycan but does not convert DslA into a form competent on acetylated material. The presence of Bd1413 (DslB) and Bd1440 (DslC) in *B. bacteriovorus* HD100 raised the question as to whether they would display similar specificity. We were able to produce these enzymes in a similar manner to DslA,

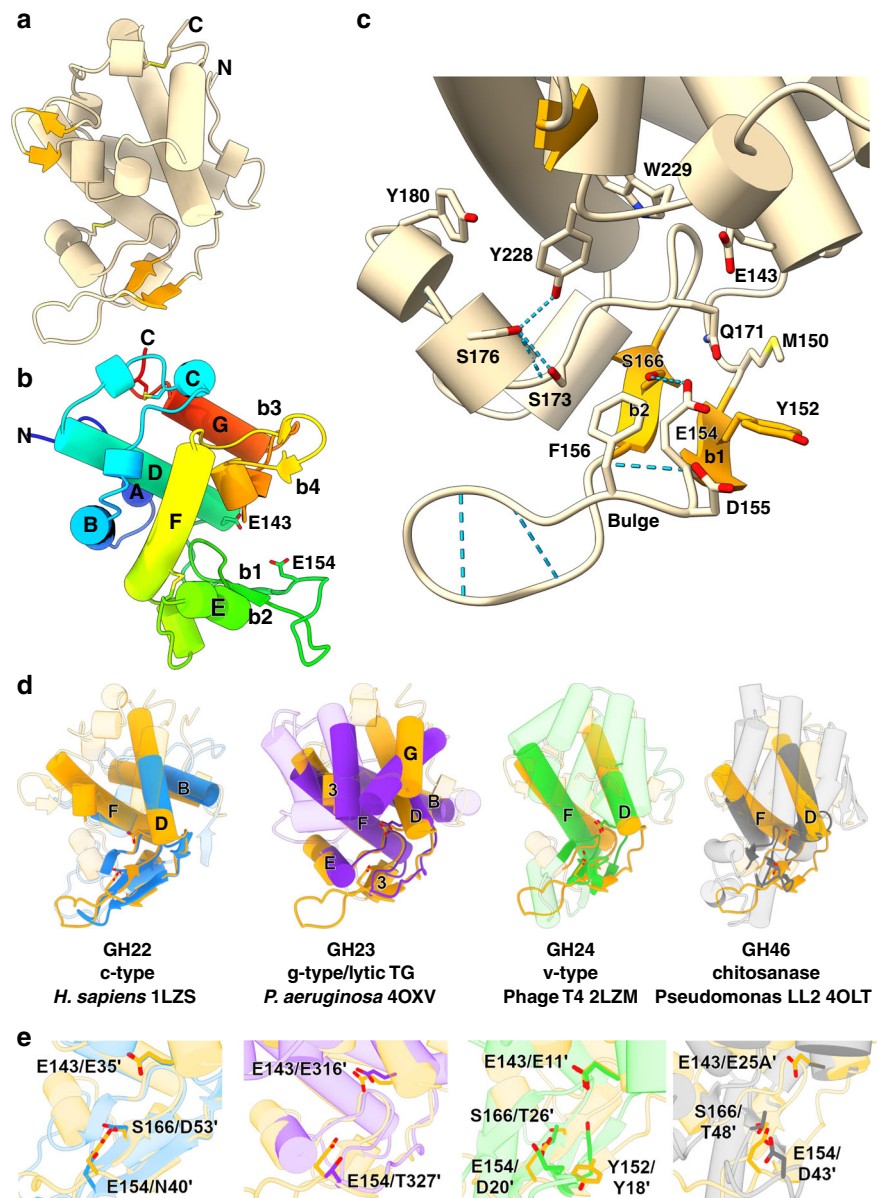

**Fig. 3 Structure of DslA. a** Fold of catalytic region of DslA (R74 to terminal K254), N and C termini labeled, and disulfide bonds presented in stick form. **b** View ~90° from **a**, protein chain colored in rainbow progression from N terminus (blue) to C terminus (red), with secondary structure labeled (standard helices via letters A–G; unlabeled helices have been defined as smaller $3_{10}$ helices, β-strands by numbers b1–4). The two acidic residues identified as participating in catalysis are presented in stick form. **c** Active site of DslA, with selected residues in stick form and hydrogen bonds as dashed blue lines. **d** Comparison between DslA (orange) and related lysozymes and glycoside hydrolases. The active-site floor β-sheet and any helices that match are represented in solid form (with DslA helix designation labeled, lettering includes "3" for the smaller $3_{10}$ helices), with non-matching elements rendered transparently. GH numbering refers to CAZy nomenclature[50], and given examples have organism, PDB code and catalytic residues noted below the fold. Family GH23 demonstrates the strongest similarity to DslA, with just the core elements (helices D, F, β-sheet) matching for v-type lysozymes and chitosanases. **e** Zoomed-in active sites from **d**, with select active-site residues labeled (DslA numbered, other fold residues noted by a prime ').

and the assay results clearly demonstrate that both DslB and DslC are bone-fide GlcNAc deacetylation-specific lysozymes (Fig. 4, righthand grouping). So although the gene for DslA was most highly expressed at the end of predation, the DslB and C enzymes were also present, possibly in reserve.

**Premature release of predator cells from bdelloplasts by exogenous addition of DslA.** As DslA showed specific activity toward GlcNAc-deacetylated peptidoglycan in vitro (Fig. 4), we wished to show its activity in vivo. We knew from our previous work[15] that GlcNAc deacetylases of *B. bacteriovorus* act on the prey PG during the first 30 min of predation. To test for DslA

activity, invaded *E. coli* prey bdelloplasts post GlcNAc deacetylation (1 h after predation was started) were used in test treatments with EDTA (to assist enzyme access through the outer membrane) and exogenously added DslA. This 1 h timepoint is early in the 4 h predation period when usually no predator cells naturally escape the bdelloplast, but when the GlcNAc of the prey peptidoglycan is already deacetylated. Additional controls were performed using EDTA plus DslA with one of the catalytic site residues inactivated (E143Q), or using conventional hen egg-white lysozyme (HEWL) or buffer only. An outline of the premature release experiment is shown in Fig. 5a. Images of the treated bdelloplasts (examples shown in Supplementary Fig. 7)

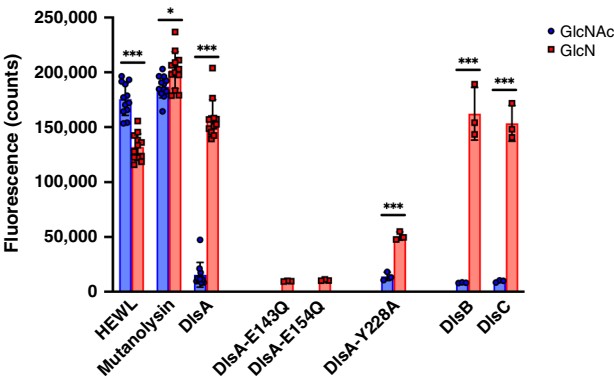

**Fig. 4 Enzyme assay of DslA, related enzymes, and general lysozymes.**
Enzymes were incubated with acetylated peptidoglycan (GlcNAc, blue) and peptidoglycan partly deacetylated via treatment with *Bdellovibrio* Bd0468 and Bd3279 (GlcN, red). Fluorescence reading monitors the hydrolytic release of FITC-labeled peptidoglycan fragments into solution. Left grouping: comparison of specificity of DslA, hen egg-white lysozyme (HEWL), and *Streptomyces globisporus* mutanolysin. The non-specific mutanolysin hydrolyzes both acetylated- and deacetylated peptidoglycan equally, but HEWL is specific for the acetylated form (with residual activity on GlcN sample indicative of mixed material, partial *N*-deacetylation[15]). DslA displays strong activity for the deacetylated GlcN sample and is unable to turnover acetylated GlcNAc peptidoglycan. Middle grouping: relative low activity of DslA putative catalytic E to Q mutants, and reduced activity for Y228A variant. Right grouping: homologous DslB and DslC proteins display a similar preference for a deacetylated substrate. Significance analysis uses two-tailed *t*-test, ***$p < 0.0005$, *$p < 0.05$. $n = 12$ independent samples for HEWL, Mutanolysin, and DslA experiments. $n = 3$ independent samples for DslA mutants and homologs (DslB and DslC) experiments. Data are represented as the mean ± standard deviation.

were categorized manually based on the different detected outcomes (Fig. 5b and Supplementary Fig. 8) and then grouped into superordinate categories dependent on the internal or external position of the *Bdellovibrio* with respect to the bdelloplast before (Supplementary Fig. 9) and after treatment (Fig. 5c). The percentage of prematurely released, external predator cells increased significantly when early bdelloplasts were treated by exogenous addition of DslA, whereas no significant change was detected for the treatment controls DslA E143Q, HEWL or buffer only (Fig. 5c). This finding was accompanied by a concomitant increase in predatory cells, that appears to be just in the process of being released (Fig. 5c, *Bdellovibrio* interm.).

Thus, the wild-type DslA, which is normally expressed at the end of predation, could act, when added exogenously with EDTA, on the GlcNAc-deacetylated PG of *E. coli* prey bdelloplasts and release predators prematurely, as would normally occur naturally later by endogenous predator expression of DslA at the exit stage of predation.

## Discussion

Starting from a theoretical/practical requirement of *B. bacteriovorus* to utilize at least one specialized "exit enzyme", our phenotypic, biochemical and structural experimentation identifies and confirms DslA as a true GlcNAc deacetylation-specific lysozyme with a structure different to other lysozyme superfamily members. In doing so, we have likely identified and structurally defined the primary active agent of the predatory culture concentrate that Ruby and co-authors used in cell extracts, in seminal early work, to prematurely release cells when studying *Bdellovibrio* differentiation[21]. The usage by *B. bacteriovorus* of

peptidoglycan GlcNAc deacetylation to mark prey versus self is a self-protection mechanism distinct from that previously observed (immunity protein inhibition of effector enzymes[13]). Our work here adds to the known "tools" of the remarkable kit of self-protection which *B. bacteriovorus* developed through evolution to be able to prey upon and destroy bacteria made of very similar chemical components to itself.

Peptidoglycan deacetylation is widely utilized in human and animal host adaptations by Gram-positive bacteria but not typically in Gram-negatives[7], which may suit its usage by the predator of Gram-negatives *B. bacteriovorus*. However, any prey cell-possessing enzymes acting downstream of deacetylation could potentially autolyze upon invasion or due to exposure to a PG lytic enzyme of self or external predator. Consistent with this idea, it is likely the lipobox of the protein acts to display and retain DslA at the outer leaflet of the predator outer membrane, acting to locally disrupt the prey's bdelloplast wall when predator progeny exit. This would be different if DslA were to be generally secreted on a signal peptide, with the potential for deleterious lysis of neighboring predators yet to complete the intraperiplasmic stage of the lifecycle (or neighboring and as yet uninvaded prey, depending on their PG acetylation status). Thus, the structure and enzymatic mechanism (discussed below), plus the likely tethering of DslA to predator are both important features for efficient predation.

The architecture of DslA is related to, but significantly different from, other members of the lysozyme superfamily (Supplementary Fig. 11) but still fulfil the inclusion criteria for the lysozyme superfamily noted by Monzingo et al. — conservation of the β-sheet floor and helices D and F, with appropriate placement of acidic catalytic residues[19]. It is interesting that DslA is structurally more similar to GH23 lytic transglycosylases than GH46 chitosanases — hence, it is probably "easier" for evolution to adapt for sugar selection than muropeptide selection. The structural equivalence of DslA E154 to the β1 secondary catalytic acid residues of goose-type lysozymes (also GH23, but with a second catalytic residue lacking in the lytic transglycosylases), v-type lysozymes and chitosanases, and our assay on the E143Q and E154Q mutants argues for an inverting hydrolytic mechanism for DslA. Residue E143 of the conserved ES motif would act as a general acid, protonating the equatorial linkage between MurNAc and GlcN, with E154 acting as a general base to activate water to attack the MurNAc C1 in an axial orientation[19]. This mechanism is distinct from HEWL and other c-type lysozymes wherein positioning of the second catalytic residue on β2 (occupied by Ser166 in DslA, Fig. 3a) results in no space for a water molecule under the substrate and a retaining mechanism[22].

One of the adaptations of DslA is to utilize a double Glu mechanism — all characterized lysozymes to date use a Glu:Asp pair. The Glu:Glu usage is however conserved in a select group of chitosanases[23], raising the intriguing possibility that a second Glu could be a favorable feature of deacetylated GlcN-metabolizing enzymes. The assay data confirm a clear specificity of DslA for GlcNAc-deacetylated peptidoglycan — our structure allows us to explain how this is achieved using a variant on the lysozyme superfamily fold. Retention of the ES motif and features surrounding the active-site cleft suggest that it is valid to assume DslA will utilize a similar (invariant) hexasaccharide binding mode to that of other lysozymes. Despite the relatively more extended β-strand features of DslA, the cleft shape and charge is not significantly different from c-type lysozymes (Fig. 6a). Hence, despite GlcN having a p$K_a$ of ~6.5, and thus GlcNAc-deacetylated peptidoglycan possessing a more positive charge than the acetylated form, DslA does not use negatively charged residues to complex the substrate. Using a GH22 chitohexaose complex (1LZS)[18] and helix D, and the ES motif as the basis of superimposition, it is possible to model a DslA substrate-binding mode

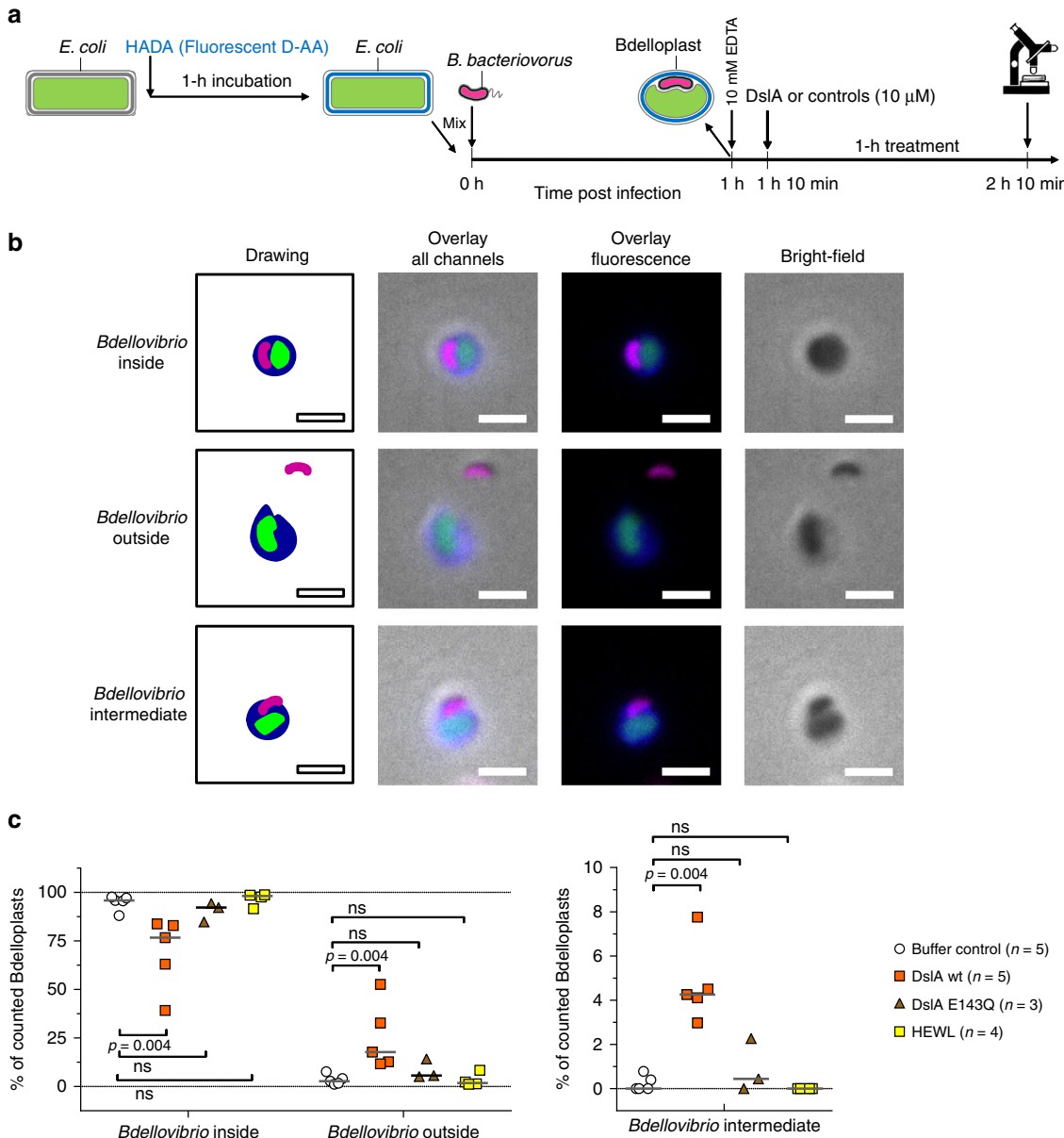

**Fig. 5 Exogenously added lysozyme DslA releases *B. bacteriovorus* from 1h bdelloplast. a** Experimental scheme including different stains/fluorescent proteins to enhance visualization of different compartments. The experimental outline is described in the legend of Supplementary Fig. 8. **b** Example images of different outcomes after treatment copied from Supplementary Fig. 8. *B. bacteriovorus bd0064:mcherry* is shown in magenta, prey peptidoglycan in blue. Scale bar is 2 μm. **c** Percentage of bdelloplasts categorized according to the position of the *Bdellovibrio* in respect to the bdelloplast after treatment as described above (Supplementary Fig. 8). Individual symbols represent % of all categorized bdelloplasts per experiment. A minimum of 71 bdelloplasts per treatment condition and biological repeat were evaluated (for details please refer to Source Data file). The number of biological repeats is indicated in the legend as *n* for the respective treatment. Horizontal lines represent median, statistical test was Mann–Whitney, ns non-significant, *p*-values = 0.004 (all *p*-values are one-tailed). HEWL hen egg-white lysozyme.

(we were unsuccessful with attempts to co-crystallize or soak various glucosamine polymers into all crystal forms). The DslA:substrate model (Fig. 6c, additional complementary models from other substrate complexes shown in Supplementary Fig. 12) are informative on several levels — GlcNAc deacetylation is preferred/utilized over MurNAc deacetylation in the Bd0468:Bd3279:DslA system because the lysozyme fold chiefly recognizes more GlcNAc 2′ units than MurNAc 2′ units, a feature resulting from the bulky 3′ muropeptide substituent of MurNAc requiring placement outside the cleft. Of the six saccharide rings, GlcNAc "c" and "e" sit with the 2′-NAc groups facing inward. These two positions are occupied in DslA by Y228 and M150, the former

placing the sidechain OH in the same position as an acetyl oxygen. Hence, specificity of DslA for GlcN over GlcNAc presumably arises via steric blockage of ring c and e if the sugar groups are acetylated. The positioning and absolute conservation of Y228 appears to be more important than the more variable M150 (Supplementary Fig. 13), and is of interest when comparing to the wider lysozyme superfamily. Residues in the vicinity of M150, including the partly conserved Y152, may assist in collectively creating an environment that favors deacetylated substrate. The greater degree of conservation at Y228 rather than M150 is in agreement with the analysis of the buried-surface area of lysozyme:substrate complexes wherein more contacts are made

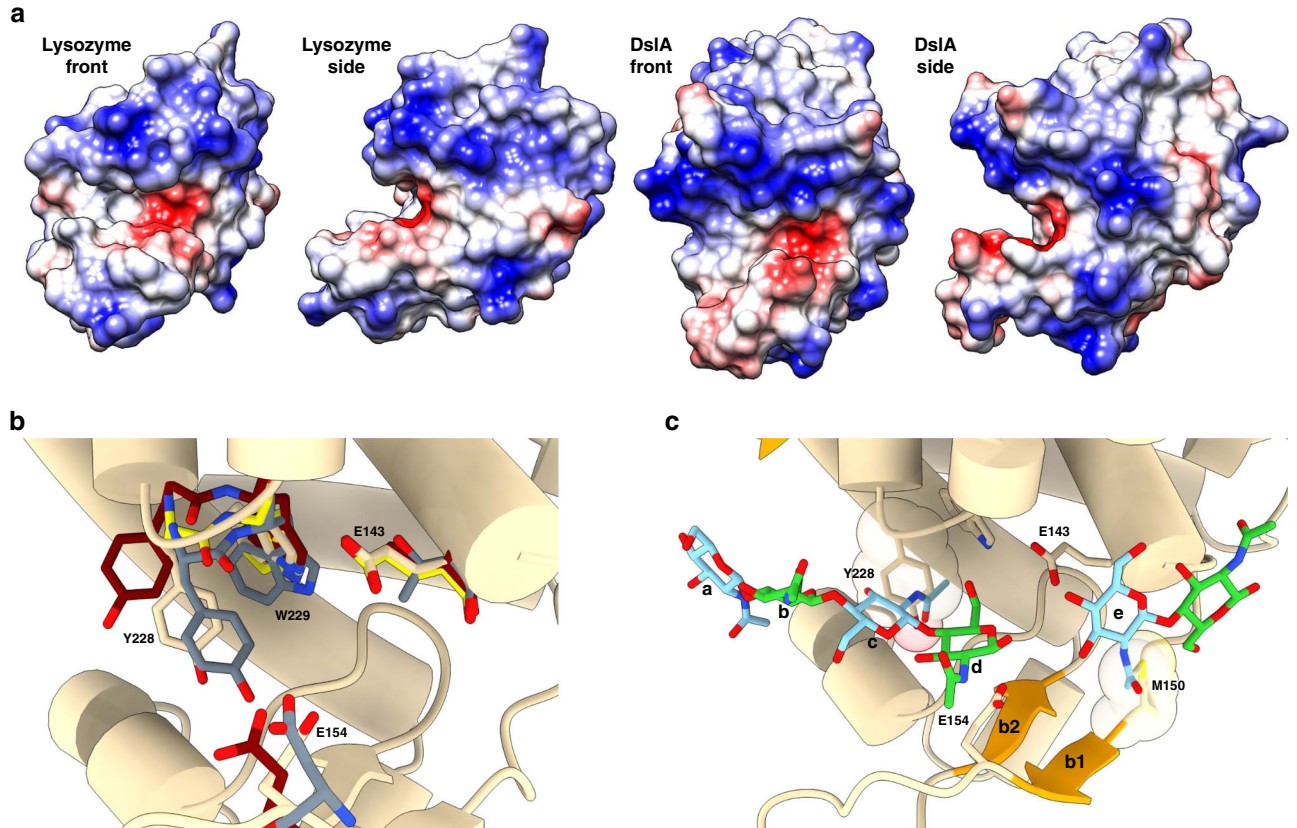

**Fig. 6 Active-site cleft and substrate-binding mode. a** Surface representation of *H. sapiens* c-type lysozyme (1LZS) and *B. bacteriovorus* DslA, with alternating views of active-site cleft. The protein surfaces (1LZS calculated without ligand) have been colored to demonstrate charge using coulombic methods, −10 to +10 kcal/(mol·e), red to blue. The two enzymes demonstrate a similar patterning at the active-site cleft, despite DslA binding deacetylated peptidoglycan. **b** Comparison of DslA to non-catalytic members of the lysozyme superfamily. The fold of DslA and residues E143, E154, and Y228-W229 are shown overlaid with equivalent residues from mouse α-lactalbumin (1NF5, gray)[24], mouse sperm lysozyme-like protein (4YF2, maroon)[25], and the active enzyme *H. sapiens* lysozyme (1LZS, yellow)[18]. The YW motif on an active-site loop of DslA is shared with α-lactalbumin and sperm lysozyme-like protein and is equivalent to an AW motif in *H. sapiens* lysozyme. The D helix catalytic residue of DslA, E143, is replaced by Thr in the two inactive enzymes, but the β-sheet catalytic residue, E154, is retained as either a Glu or an Asp. **c** DslA superimposed with substrate from a liganded form of lysozyme (1LZS). The classic "4 + 2" lysozyme saccharide binding pose is shown, with the ligand coordinates from 1LZS superimposed into DslA using the ES motif of both enzymes as a guide. The cell-wall sugar rings are labeled **a**–**f**, with GlcNAc and MurNAc residues colored blue and green, respectively. The 2′ N-acetyl group of rings c and e face inwardly into the rear face of the active-site cleft and provide a model for deacetylation specificity: the sidechains of residues Y228 and M150 (vdw surface shown in transparent) sterically occupy the same space as the acetyl group atoms.

to ring c (−2 position) rather than ring e (+1). Hence, DslA is also likely to follow this pattern, discriminating most strongly via the central region of the active-site cleft.

We were able to find two structurally characterized wider superfamily members that retain a YW motif at the location of DslA Y228:W229; most lysozymes have a smaller residue preceding the tryptophan (e.g., AW in 1LZS, Fig. 6b). Interestingly, these two YW variants, α-lactalbumin[24] and mouse sperm lysozyme-like protein[25], are both annotated as inactive enzymes. Hence, like DslA, these proteins may use the YW motif to block binding of acetylated GlcNAc peptidoglycan. Mouse sperm lysozyme-like protein and α-lactalbumin replace the first catalytic residue equivalent to E143 with a catalytically incompetent Thr, but retain an equivalent to the second acidic residue DslA E154 (Fig. 6b). We observe a lesser activity for the Y228A variant of DslA (Fig. 4), but this does not "convert" to an enzyme competent in turning over acetylated peptidoglycan — thus, substrate selection is likely to involve additional factors to steric block by Y228, possibly selecting for the subtly different torsion angles between GlcNAc and GlcN or other differences in recognition. One of these factors may be the topology of the peptide backbone in this region; the carbonyl of Y228 projects out relatively further

than equivalent residues in the superfamily, and mutation to alanine will not nullify this feature. Factors additional to Y228 that could potentially contribute to specificity may be complex and beyond the scope of this study (e.g., active-site dynamics, or enzyme/substrate hydration differences in relation to other superfamily members). Chitosanases often use tyrosine to recognize the amine group of GlcN[26], and the local hydrogen-bonding network of DslA Y228-S176-S188 looks to be a suitable pocket for recognition of ring c GlcN.

Our biochemical observation of a similar GlcNAc deacetylation-specificity for DslB and DslC (30% identity to DslA and encoded as a back-up similar to other duplicated predatory enzyme genes of *B. bacteriovorus* as for DacBs[12] and GlcNAc deacetylases[15]) gives confidence to the process of identifying Dsl homologs via sequence comparison. Using the DslA sequence, we were able to find homologs in several other prey-invasive *Bdellovibrio* strains (60–70% identity). However, in line with invasive predatory association and replication of *B. bacteriovorus* within the prey cell wall, we did not find homologs in the epibiotic predators that have no requirement for an exit enzyme (a search of the genomes of the model epibiotic predators *B. exovorus* and *Micavibrio aeruginosavorus* revealed no DslABC homologs (no

hits with an *E*-value < 0.1)). A range of sequences from diverse bacteria were obtained that have 24–43% sequence identity (Supplementary Fig. 13), and can be validated in that all sequences possessing an equivalent of E154 also have absolute conservation of Y228. Several of these strains have identifiable CE4 deacetylase genes, and so will be able to generate the GlcN cell-wall substrate. Hence, it will be interesting to see if these systems use coupled GlcNAc deacetylation and GlcN-hydrolysis in a biologically important manner. An in-depth analysis of the roles of DslB and DslC, which were not differentially expressed at exit time in *B. bacteriovorus*, is beyond the scope of this study. However, we did show that *B. bacteriovorus* strains with mutated/deleted DslB or DslC showed small but not significant delays in prey entry or exit compared to wild type (Fig. 2 and Supplementary Fig. 3), thus they potentially represent a spare, secondary, enzyme set. Gene duplication giving rise to pairs of enzymes contributing to a single PG-modifying task has been previously demonstrated for D,D-endopeptidases[12] and N-deacetylases[15] in *B. bacteriovorus*. As we found in a previous study that 50% of GlcNAc-deactylase double deletion mutant strains get "trapped" in the outer periplasm between prey wall and outer membrane[15], there may be a requirement for a localized Dsl-activity during entry of predators to prey, as prey wall deacetylation is occurring at the time of invasion. (Other enzymes are also involved in the multi-factorial invasion process[14] and are the subject of a current extensive experimental program beyond this study.) The majority of the lysozyme-like activity on prey entry seems to be from Bd1411, with a potential minor contribution by DslA (Supplementary Fig. 3). Although predominant gene expression (Fig. 1) and DslA activity is in the exit phase (Fig. 2), it is possible that some DslA enzyme originating from the exit remains attached to the *Bdellovibrio* outer membrane, as it is a lipoprotein. In this way DslA could persist after exit and potentially still be used at entry, which will be the subject of another multi-enzyme study.

The fact that the ΔdslA mutant did ultimately leave the prey cells after a delay (Fig. 2) indicates that additional predatory enzymes are involved in the overall exit process which does involve breakage of the whole outer envelope, including wall but also an outer membrane of the prey bdelloplast. There was no obvious difference in the way by which ΔdslA mutant predator progeny break the prey cell into remnants on exit in comparison to the wild-type exit mode (Supplementary Fig. 14). This contrasts the two different exit modes observed for the double GlcNAc-deactylase mutant where some whole prey cell-sized "ghost" remnants were seen in cultures after predation[15]. This can be explained by the different timepoints respective enzymes are active within the predatory lifecycle and the outcomes of them being missing in *B. bacteriovorus* mutants. GlcNAc-deacetylase enzymes were previously found to act early in predation at invasion time and their deletion in a double Δbd0468 bd3279 *B. bacteriovorus* mutant caused approximately half of the predators to fail to enter through the prey cell wall, instead carrying out predation in the outer periplasm, between the wall and outer membrane, and approximately half entering the inner periplasm. The outer dwelling half of the Δbd0468 bd3279 deacetylase mutant population left behind intact prey cell "ghosts" of wall and membranes (seen by fluorescently labeled prey cytoplasm being detected in osmotically stable spheres) after predation. The inner dwelling half of the population exited through the original pore by which they had traversed the prey cell wall — again leaving behind smaller prey remnants. In contrast, the ΔdslA mutant has a different phenotype (Supplementary Fig. 15) and the population predominantly cross the prey wall, likely due to the ΔdslA mutant still expressing active Bd1411, and grow in the inner periplasm (expression of *dslA* being higher after prey invasion), without leaving extensive prey remnants.

Our work here identifies DslA as important to *B. bacteriovorus* predatory biology, and confirms that evolution has shaped a divergent lysozyme homolog family to be specific for GlcNAc-deacetylated peptidoglycan, and that, importantly, this enzyme contributes to prey cell exit by predatory bacteria. The duplication and diversification of known cell wall-metabolizing activities for a "new" purpose in the predatory *B. bacteriovorus* lifecycle is a theme observed before for D,D-endopeptidases[12] and L,D-trans-peptidases[14], and can be inferred for other activities also[27]. DslA (or homologs), if adapted for Gram-positive peptidoglycan may find use as a tool in the study of such pathogens that deacetylate their wall to escape conventional lysozyme activity[4]. Our observations will assist in unraveling the complexity of *Bdellovibrio* prey cell manipulation and provide a basis for identifying the biological role of cryptic homologs of DslA.

## Methods

**Growth of bacterial strains**. *Bdellovibrio bacteriovorus* strain HD100[T] was used throughout, and wild type, or mutated Δbd0314 (ΔdslA), Δbd0314::pK18-bd0314, Δbd1411, Δbd1440 (ΔdslC), bd1413::pK18-bd1413EQmCherry (bd1413EQmc), or bd0064:mcherry strains were maintained on Ca/HEPES buffer (25 mM HEPES, 2 mM CaCl₂, pH 7.6) with late-log phase *E. coli* S17-1 prey (if not indicated differently) at 29 °C for 16–24 h with 200 rpm shaking[28–30]. Various *E. coli* strains were grown in YT broth at 37 °C and 200 rpm plus additional substances as indicated, if required.

**RNA isolation from predatory lifecycle/HI grown cells and RT-PCR analysis**. Synchronous predatory infections of *B. bacteriovorus* HD100 on prey *E. coli* S17-1 were set up by concentrating 1 l predatory cultures by centrifugation at $16,900 \times g$ and resuspending in 100 ml Ca/HEPES buffer, then mixing 50 ml of this with 30 ml Ca/HEPES buffer and 40 ml *E. coli* S17-1 back diluted to OD₆₀₀ 1.0 in Ca/HEPES buffer[31]. Controls of predator only and prey only were also set up. HI stains were grown up to an OD₆₀₀ₙₘ of 0.6 and all samples were treated as with 1% phenol, 19% ethanol for 45 min at 4 °C, before centrifugation at $5000 \times g$ for 10 min and the pellet stored at −80 °C (ref. [31]). RNA was isolated from these pellets using the Promega SV total RNA isolation kit with RNA quality being verified by an Agilent Bioanalyser using the RNA Nano kit. All samples for Supplementary Fig. 2 with the RT-PCR on HI strains were matched to 10 ng μl⁻¹ RNA prior to RT-PCR. The Qiagen One-step RT-PCR kit was used to perform the following reaction conditions: one cycle 50 °C for 30 min, 94 °C for 15 min, then 25–30 amplification cycles of 94 °C for 1 min, 50 °C for 1 min, 72 °C for 1 min, and finally a 10 min extension at 72 °C after the last cycle, followed by a 4 °C hold. All experiments to monitor transcription during HD lifecycle were carried out with at least two biological repeats. RT-PCR primers to amplify an ~100-bp-long fragment were 5′-TATCGA ACCGGAAGTCGTTC-3′ and 5′-GCAACCTTGTCTTCCCAAAG-3′ for bd0314 (dslA), 5′-ACAAACAGAAGGACCGCAAG-3′ and 5′-TCATCACATGACGACCC AAC-3′ for bd1411, 5′-CTCCAGAGCCTGAACCAGAG-3′ and 5′-CCTGCAGG TATTTGGACCAC-3′ for bd1413 (dslB), 5′-TGGGAAACTTCCACAAATCC-3′ and 5′-GGCAGAACTTGCTCATGTCA-3′ for bd1440 (dslC), and 5′-TGAGGAC GAGATCAAACGTG-3′ and 5′-AAACCAGGTTGTCGAGGTTG-3′ for dnaK (bd1298).

**Generation and complementation of lysozyme mutants in *B. bacteriovorus* HD100**. Markerless deletion mutants ΔbdO314 (ΔdslA) and Δbd1411 of the bd0314 and bd1411 open reading frames in *B. bacteriovorus* HD100 were generated using a modified version of Steyert and Pineiro[32] by subcloning 1 kb of up- and downstream flanking DNA first in pUC19, then excising this fragment and cloning into the suicide vector pK18mobsacB[33], which was then transformed into *E. coli* S17-1 for conjugating into wild-type *B. bacteriovorus* HD100. Conjugations were carried out for 16 h at 29 °C on PY agar (10 g l⁻¹ Difco Bacto peptone, 3 g l⁻¹ Difco Yeast extract 10 g l⁻¹ agar) by concentrating 10 ml of a predatory *B. bacteriovorus* and 10 ml donor *E. coli* S17-1 by centrifugation at $5500 \times g$ for 10 min and resuspending each in 100 μl Ca/HEPES buffer and placing on a nylon membrane. Exconjugants were then grown as predatory cultures in Ca/HEPES buffer supplemented initially with 50 μg ml⁻¹ kanamycin selection, then subsequently in Ca/HEPES buffer supplemented with 5% sucrose to select for a second crossover[29,30]. The primers to generate the deletion mutant were: Bd0314-F 5′-AGCTACGCATGCACCGCCAG AACCAAAATCCG-3′; Bd0314-ΔR 5′-GCTGGTCTTGTCACCGGCACTCTAGA CCAGGATCTGAACAACAAGC-3′; Bd0314-R 5′-CTTAGCGGATCCGGCGAT GATATCAGAACCAT-3′; Bd0314-ΔF 5′-GCTTGTTGTTCAGATCCTGGTCTAG AGTGCCGGTGACAAGACCAGC-3′; Bd1411-F 5′-TGCTAGAAGCTTGTTGTG AATCACTCATTGCC-3′; Bd1411-ΔR 5′-CGCCTGATCTTACTTGCAGCATGC AGCAAGCATGATGCCGGA-3′; Bd1411-R 5′-GACTAGTCTAGATCTCTGGTG

AGATCAACACG-3′; and Bd1411-ΔF 5′-TCCGGCATCATGCTTGCTGCATGCT GCAAGTAAGATCAGGCG-3′.

Markerless deletion mutant Δbd1440 (ΔdslC) was generated by assembling 220 bp upstream region and first 6 bp of open reading frame bd1440 with the last 9 bp of the reading frame with 299 bp downstream flanking DNA. For Gibson assembly with the NEBuilder kit (New England Biolabs) the following primers where used: Bd1440-F 5′-CGTTGTAAAACGACGGCCAGTGCCAATCCCCGGCCATCAGC CA-3′; Bd1440-ΔR 5′-GTTACTTACACGTCAAAAGACCTCCAAAGTGTC-3′; Bd1440-ΔF 5′-TCTTTTGACGTGTAAGTAACAAGACGGTCGGG-3′, and Bd14 40-R 5′-GGAAACAGCTATGACCATGATTACGGTCTTCAAAGTTGTCCCG-3′. Complementation of Δbd0314 was achieved by single crossover with plasmid pK18-bd0314 containing a kanamycin resistance. Plasmid pK18-bd0314 was conjugated into B. bacteriovorus Δbd0314 from donor E. coli strain S17-1 as described above. Plasmid pK18-bd0314 containing the bd0314 (wild type) open reading frame with 1000-bp upstream and 61-bp downstream region on conjugable vector pK18mobsacB[33] was generated by Gibson assembly with the NEBuilder kit (New England Biolabs) with primers Bd0314Comp-F 5′-CGTTGTAAAACGACGGCCAGTGCCAATCGCCTT GTTTGCGACAATTG-3′ and Bd0314Comp-R 5′-GGAAACAGCTATGACCATGA TTACGGGATGCTCCTTTTGTTTAAGCG-3′ using gDNA of B. bacteriovorus HD100 as a template. Strain bd1413::pK18-bd1413EQmCherry (bd1413EQmc) was obtained by a single crossover with plasmid pK18-bd1413EQmCherry containing a kanamycin resistance. Plasmid pK18-bd1413EQmCherry was conjugated into B. bacteriovorus HD100 wild type from donor E. coli strain S17-1 as described above. Plasmid pK18-bd1413EQmCherry for the inactivation of the active-site E151 (to Q151) of Bd1413 (DslB) in was achieved by mutagenic PCR using the primers Bd1413mCherryF 5′-GTACTGGAATTCATGATCGCAACAAAGAACCA-3′, 1413 FOE 5′-CGCCAAATATCAGAGCGCCTACAG-3′, Bd1413mCherryR 5′-TGACGA GGTACCTTTGCACAAGGGAAGAGTCT-3′ and 1413ROE 5′-CTGTAGGCGCTC TGATATTTGGCG-3′. The resulting PCR product was subcloned into pAKF56 (named pMAL_p2-mCherry in Fenton et al.[34]) with EcoRI and KpnI to fuse mCherry with the mutated bd1413. This was then cloned into the conjugable vector pK18mobsacB with EcoRI and HindIII.

**Time-lapse microscopy to examine entry time.** Phase-contrast time-lapse microscopy was carried out on predation of E. coli S17-1 prey by B. bacteriovorus HD100 wild type, Δbd0314, Δbd1411, and Δbd1440. B. bacteriovorus strain bd1413EQmc was tested by predation on E. coli S17-1:pZMR100 (ref. [31]) in Ca/HEPES buffer with kanamycin. Fifty times concentrated 24 h incubated predatory Bdellovibrio culture was mixed with an equal volume of 5× concentrated late-log phase E. coli prey by centrifugation (17,000×g, 2 min, room temperature). Immediately after mixing, 10 μl of the mixture was applied onto a 0.3% agarose in Ca/HEPES buffer pad on a microscope slide for phase-contrast time-lapse microscopy. Imaging was performed at ambient temperature with the Nikon Eclipse E600 microscope using a ×100 objective lens (numerical aperture (NA), 1.25), and an exposure time of 0.01 s (gain 0). Images were acquired using a Hamamatsu Orca ER camera and Simple PCI software (version 5.5). Sequential imaging every 1 min for six fields of view was enabled by an H101A xy motorized stage (Prior Scientific, minimum step size, 0.01 μm), and a frictional z axis controller (minimum step size, 2 nm) in conjunction with the Simple PCI software, which enabled fine auto-focusing. Measurements are from at least three independent biological replicates with 90 events evaluated per predatory strain.

**Time-lapse microscopy to determine exit time.** 24-h incubated cultures of B. bacteriovorus HD100 wild type, Δbd0314 or Δbd1440 growing predatorily on a late-log phase E. coli S17-1 culture in Ca/HEPES buffer at 29 °C and 200 rpm were filtered through a 0.45 μm filter and concentrated 50× by centrifugation (3877 × g for 20 min at 29 °C). The same procedure was performed with B. bacteriovorus bd1413::pK18-bd1413EQmCherry (bd1413EQmc) or Δbd0314::pK18-bd0314 pregrown on E. coli S17-1:pZMR100 (ref. [31]) prey in Ca/HEPES buffer with 50 μg ml$^{-1}$ kanamycin. However, to minimize kanamycin carry over Bdellovibrio cells were washed once with Ca/HEPES buffer containing no antibiotics (additional 5-min spin at 3877 × g and 29 °C). Each culture of B. bacteriovorus was mixed with an equal amount of E. coli S17-1::pMAL-p2_mCherry[34] (grown overnight in YT with 50 μg ml$^{-1}$ ampicillin and 200 μg ml$^{-1}$ IPTG at 37 °C and 200 rpm, concentrated to OD$_{600nm}$ 5.0 and washed twice in Ca/HEPES buffer). This semi-synchronous infection was incubated at 29 °C and 200 rpm for 2 h 45 min before a 10 μl sample was transferred onto a 1% agarose in Ca/HEPES buffer pad on a microscope slide for time-lapse microscopy. Time-lapse epifluorescence microscopy was performed using the same Nikon Eclipse E600 microscope system as described for the entry time-lapse microscopy. Exposure time was 0.1 s for bright field (gain 0) and 4.1 ms (gain 255) for hcRED filter block, and eight fields of view were imaged sequentially every 2.5 min per experiment. Measurement of prey exit times — defined as the time between predator progeny separation (estimated frame in which the predator spiral cellular filament structure divides to form separate progeny) and first escape from prey cell remnants (first frame in which a predator progeny cell visibly starts to leave the prey structure and subsequently moves out by at least half a progeny cell length) — was from at least two biological repeats per strain. Only bdelloplasts were evaluated, which were reasonably in focus, fluorescence was bright enough and B. bacteriovorus progeny cells had started to exit up to 280 min after starting

time-lapse microscopy (n = 45 for each strain, except for Δbd1440, where n = 21). All Serial images were visually inspected using Fiji/ImageJ (1.52i)[35,36].

**Determining the number of prey residual structures per E. coli.** Cultures of B. bacteriovorus HD100 wild type or Δbd0314 were grown on late-log phase E. coli K-12 MG1655 in Ca/HEPES buffer at 29 °C and 200 rpm for 24 h. These predatory cultures were filtered (0.45 μm), concentrated 10× by centrifugation (3877×g for 20 min at 29 °C) and then incubated at least 10 min at 29 °C and 200 rpm before infection of prey. An overnight prey culture of E. coli K-12 MG1655 (grown in YT at 37 °C and 200 rpm) was filtered through an autoclaved Millex®-AP filter (Prefilter 25MM, Millex-AP glass fiber membrane, Merck) to remove any aggregated/elongated prey cells. The filtered E. coli K-12 MG1655 culture was diluted to an OD$_{600nm}$ 1.0 with Ca/HEPES buffer, spun for 2 min at 17,000×g and resuspended in the same volume of Ca/HEPES buffer. A semi-synchronous predatory infection was set up by mixing 0.3 ml Ca/HEPES buffer with 0.4 ml E. coli K-12 MG1655 (OD = 1.0) and 0.5 ml B. bacteriovorus HD100 wild type or Δbd0314 (10× concentrated), and incubated at 29 °C and 200 rpm. Prey remnants were stained 5 h post-infection with Film tracer$^{TM}$ FM$^{TM}$ 1–43 green biofilm cell stain used to stain lipids (final concentration 5 μg ml$^{-1}$, ThermoFisher Scientific) and peptidoglycan stained with Wheat Germ Agglutinin (WGA)-Alexa Fluor® 350 conjugate (final concentration 10 μg ml$^{-1}$, ThermoFisher Scientific). After stain addition, samples were incubated for 15 min in the dark. For determination of the number of bdelloplasts, samples were spiked with latex beads (polystyrene, 0.3 μm mean particle size, Sigma) at 1 h and 5 h post-infection. Samples were imaged using a Nikon Ti-E inverted fluorescence microscope equipped with Plan Apo λ 100×/1.45 Ph3 objective, an Andor Neo sCMOS camera, a FITC filter cube and NIS-Elements AR software (version 4.6). Samples were viewed and imaged in phase contrast, in the DAPI channel (Ex: 395 nm/Em: 435 nm) to detect the WGA-Alexa Fluor 350 conjugate-stained prey peptidoglycan remnants and in the GFP channel (Ex: 470 nm/Em: 515 nm) to view membranes stained with FM 1–43. Exposure time was 142 ms for phase contrast (at gain 1, 12-bit), 2 s for DAPI channel and 70 ms for GFP channel (both acquisitions at gain 4, 12-bit). Ten images per strain (wild type or Δbd0314) and timepoint (1- or 5-h post-infection) were processed using Fiji/ImageJ software (1.52i)[35,36]. All images were cropped to a size of 110.03 μm × 45.30 μm visually selecting for the parts most in focus. These images were used for bead, bdelloplast and prey cell remnant counting. Detection and counting of prey cell remnants was performed on the fluorescence images obtained in the DAPI channel in Fiji PlugIn MicrobeJ (version 5.11z)[37]. Latex beads and bdelloplasts (for the 1-h post-infection samples only) were counted manually using the Cell Counter function in Fiji/ImageJ[35,36]. The ratio of bdelloplasts (rounded prey cells infected with Bdellovibrio) to latex beads at 1 h post-infection was calculated and compared to the ratio of WGA-Alexa Fluor® 350 conjugate-stained prey remnants versus polystyrene beads at 5 h post-infection. Four independent biological repeats were evaluated.

**Cloning, expression, and purification.** Constructs of dslA (bd0314, AA 74–254), dslB (bd1413, AA 80–271), and dslC (bd1440, AA 75–266), all lacking signal peptides, lipobox, and region of presumed disorder, were amplified from B. bacteriovorus genomic DNA. The amplified genes were inserted into a modified pET28b-MBP expression plasmid (TEV-cleavable His$_6$-Maltose-Binding Protein tag) by restriction-free cloning. Mutant variants of DslA were produced by standard site-directed mutagenesis. Constructs were confirmed by sequencing, and introduced into the commercially available E. coli SHuffle T7 expression strain (NEB) in order to promote disulfide bond formation. LB media supplemented with 1% w/v glucose was used to culture expression strains, and protein production induced at OD$_{600}$ of 0.6 using 1 mM IPTG (temperature lowered to 18 °C, cells were grown overnight).

Cell pellets were resuspended with appropriate amounts of lysis buffer (50 mM HEPES pH 7.5, 20 mM imidazole, 300 mM NaCl, 0.05% w/v Tween 20) and lysed using sonication, with centrifugation at 33,000 × g (4 °C) for 1 h to clarify the lysate. The supernatant was then loaded onto HisTrap HP columns (GE Healthcare) pre-equilibrated in lysis buffer. Columns were washed with ~20 column volumes of lysis buffer and protein eluted using elution buffer (50 mM HEPES pH 7.5, 300 mM NaCl, 400 mM imidazole). TEV protease was added (1:50 ratio) to pooled elution fractions and the sample dialyzed overnight at 4 °C (in 20 mM HEPES pH 7.5, 200 mM NaCl). The cleaved protein was separated from residual fusion protein by a second passage over the HisTrap column. The resulting purified protein was concentrated to ~15 mg ml$^{-1}$ and utilized in crystallization and kinetics experiments.

**Structure determination.** Crystals were grown at 18 °C using the sitting drop technique with drops composed of equal volumes of protein and reservoir solution. For wild-type DslA, crystals were obtained in 0.1 M Na acetate pH 5.5 and 50% w/v polyethylene glycol (PEG) 400 (P2$_1$2$_1$2 form), or 0.2 M Lithium sulfate, 0.1 M Na acetate pH 4.5, and 30% w/v PEG 8000 (P2$_1$2$_1$2$_1$ form). For DslA E143Q and E154Q mutants, crystals were obtained in 0.2 M Lithium sulfate, 0.1 M Na citrate pH 5.5, and 20% w/v PEG 3000. Other than the P2$_1$2$_1$2 form (flash cooled directly), crystals were cryoprotected in mother liquor supplemented with 20% (v/v) ethylene glycol before being flash cooled in liquid nitrogen. Diffraction data were collected at the Diamond

Light Source in Oxford, UK. Data reduction and processing were completed using XDS and the *xia2* suite[38]. Initial phasing of DslA was achieved using a merged sulfur-SAD data set (9000 frames, 0.1° oscillations, using two distinct regions of the crystal to minimize radiation damage) of an E143Q crystal collected at a long wavelength of 1.77 Å. Unmerged data were input into PHENIX AutoSol[39], which located 13S sites with a FOM of 0.3, allowing iterative cycles of building and model-based phasing improvement. The other crystal forms were solved using the E143Q search model in PHASER[40]. Protein structures were built/modified using COOT[41], with cycles of refinement in PHENIX[39] and PDB-REDO[42].

**Peptidoglycan digestion assay**. Isolated peptidoglycan (*E. coli*) was labeled with FITC in a similar method to that described by Maeda et al.[43]. Briefly, ~28 mg of peptidoglycan was incubated with 8 mg of FITC in 300 μl H$_2$O for 4 h, before being washed and resuspended in deacetylation/assay buffer (20 mM HEPES pH 7.5, 300 mM NaCl). For deacetylation-specific assays, labeled peptidoglycan was incubated at 37 °C for 12 h with two *Bdellovibrio* deacetylases, Bd0468 and Bd3279 (1 μM each). All assayed enzymes (DslABC, HEWL, mutanolysin) were incubated in assay buffer, with the 60 min time course using 100 μg ml$^{-1}$ peptidoglycan and 5 ng of enzyme, and samples taken at regular intervals, with the reaction quenched by removing the insoluble substrate using a 0.1 μm filter. Fluorescence counts were measured using a PheraStar FS plate reader set to excite at 485 nm and detect at 520 nm.

**Early bdelloplast-treatment assay**. A schematic overview of this experiment is given in Fig. 5a. *B. bacteriovorus bd0064:mcherry* (*B. bacteriovorus* with a fluorescent cytoplasm)[44] instead of the wild-type strain was used to clearly distinguish between predatory cells and other components present in the assay at the end of treatment with DslA or controls. *B. bacteriovorus bd0064:mcherry* was grown for 24 h on late-log-phase *E. coli* S17-1 prey in Ca/HEPES buffer. This prey lysate culture was filtered (0.45 μm) and concentrated 20× by centrifugation (20 min at 3877×*g*, 29 °C). The protocol of Kuru et al.[14] was used as guidance to stain the prey PG with HADA. An overnight prey culture of *E. coli* S17-1 pAKF220 (ref. [45]) (expressing mNeonGreen in the cytoplasm), grown induced with 200 μg ml$^{-1}$ IPTG on YT with 50 μg ml$^{-1}$ ampicillin, was diluted to an OD$_{600nm}$ of 1.0 with YT. After pelleting the cells (2 min at 17,000×*g* at room temperature), the cells were resuspended in an equal amount of YT (without ampicillin) and 500 μM HADA[46,47] (a fluorescent D-amino acid). This mixture was incubated for 30 min at 37 °C and 200 rpm. Labeled *E. coli* S17-1 pAKF220 was washed twice (2 min at 17,000×*g* at room temperature) in a volume corresponding to OD$_{600nm}$ 1.0 in Ca/HEPES buffer. The infection was started by mixing equal volumes of concentrated *B. bacteriovorus bd0064:mcherry* with labeled *E. coli* S17-1 pAKF220 and subsequent incubation at 29 °C and 200 rpm. Semi-synchronicity of infection was confirmed by microscopic observation at 30 min post-infection. At one hour post infection 10 mM ethylenediaminetetraacetic acid (EDTA) was added to the lysate containing early bdelloplasts and the mixture was incubated for 10 min (at 29 °C, 200 rpm) to mediate access of the exogenously added enzymes. DslA, DslA E143Q, and hen egg-white lysozyme (Sigma) were added to the assay at a final concentration of 10 μM, whereas an equal volume of Ca/HEPES buffer was added for the buffer control. These treatment assays were incubated for one hour at 29 °C and 200 rpm. Just after the addition or 1 h of treatment with DslA or controls 5 μl from the assay was transferred onto a 1% agarose in Ca/HEPES buffer pad on a microscope slide. Samples were imaged using a Nikon Ti-E inverted fluorescence microscope equipped with Plan Apo λ ×100/1.45 Ph3 objective, an Andor Neo sCMOS camera, a FITC filter cube and NIS-Elements AR (version 4.6). Samples were imaged in phase contrast (exposure 4.8 ms), the DAPI channel (Ex: 395 nm/Em: 435 nm, exposure 100 ms) the GFP channel (Ex: 470 nm/Em: 515 nm, exposure 20 ms) and the mCherry channel (Ex: 555 nm/Em: 632 nm, exposure 1 s) (with all acquisitions at gain 4). Fiji/ImageJ software (1.52i)[35,36] was used for image processing. Bdelloplasts that were reasonably in focus were sorted manually by visual inspection into the different categories shown in Supplementary Fig. 8. An overview of the categorized bdelloplasts including total numbers are given in the Source Data xls spreadsheet. The percentage of each category with respect to all bdelloplasts counted under the given conditions were calculated and merged into superordinate categories based on the position of *Bdellovibrio* with respect to the PG of the bdelloplast (please refer to Source Data). The different numbers of independent biological repeats for different treatments evaluated are indicated in Fig. 5 and Supplementary Fig. 9 as *n*.

**Phylogenetic tree**. Protein sequences were aligned using ClustalW and phylogenetic trees were generated in MEGA-X software version 10.0.05, using the Maximum Likelihood method with 500 bootstraps[48]. Trees were visualized in FigTree version 1.4.4 (http://tree.bio.ed.ac.uk/software/figtree/). Additional protein homologs were found through the NCBI BLAST program suite[49].

**Reporting summary**. Further information on research design is available in the Nature Research Reporting Summary linked to this article.

## Data availability
Coordinates and structure factors have been deposited in the PDB under accession codes 6TA9 and 6TAB for the two wild-type protein crystal forms; 6TAD for the E143Q mutant; and 6TAF for the E154Q mutant. Source data are provided with this paper for

Figs. 1b, 2, 4, 5c and Supplementary Figs. 1b, 2a, S3, 5, and 9 in the Source Data file. Other data are available from the corresponding author upon request. Source data are provided with this paper.

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

## Acknowledgements
This work was funded by BBSRC BB/M010325/1 and BB/J015229/1 (to R.E.S. and A.L.L. for C.L. and R.T.), BBSRC DTP studentships (to C.H., L.J.R. and H.S.), SNSF Postdoc Mobility fellowships P2SKP3_174680 and P400PB_183896 (to S.G.H.), and BBSRC BB/S010122/1 (to P.J.M.). R.E.S. and A.L.L. are currently supported by a Wellcome Trust Investigator Award 209437/Z/17/Z.

## Author contributions
All authors contributed to experimental design, which was supervised by A.L.L. for protein biochemistry and structure determination and by R.E.S. for microbiological assays and gene expression; C.H. cloned and purified DslABC, did all the structural characterizations, and assayed enzymes in vitro with assay design and supervision by P.J.M. S.H. conducted and evaluated all the video exit phenotypes of *B. bacteriovorus* mutants with help of C.L. All mutants (except Δ*dslC*) were constructed and partially characterized by H.S. and C.L. Entry phenotypes were assayed by H.S. and S.H. with the assistance of C.L. S.H. used DslA protein produced by C.H. to perform ex vivo premature release assays; and carried out microscopy and image analysis with assistance from C.L. G.T. constructed Δ*dslC* and conducted two repeats of RT-PCR on *bd1440* with assistance from R.T. L.J.R. performed phylogenetic analysis. S.H., C.H., A.L.L., and R.E.S. wrote the manuscript, and all authors approved of the final manuscript submitted.

## Competing interests
The authors declare no competing interests.
