## [Peer Review File · Nature Communications]

Reviewers' comments:

Reviewer #1 (Remarks to the Author):

The manuscript by Harding et al describes the structural determination of lysozyme DslA from the bacterial predator *Bdellovibrio bacteriovorus*. During release of the bacterial progeny this bacterium uses deacetylases to modify the cell wall (peptidoglycan, PG) from the prey, and subsequently hydrolyze it by the use of specific lysozymes. The present work associates such lysozyme with lipoprotein DslA and presents its three-dimensional structure for the catalytic domain. The manuscript is clear and methodology sound. The structure of the wt and two mutants for the catalytic residues are provided at high-resolution revealing a fold related with that of hen egg-white lysozyme (HEWL) and GH23 family of family 1 of Lytic transglycosylases (LTs). While some bacteria modify its PG by deacetylation in order to avoid recognition by lysozymes from the host, bacterial predators such as bacteriophages or *B. bacteriovorus* have developed specific lysozymes that overcome such problem in order to release their progeny. The manuscript is well written and all experiments seem to have been performed well. However, this reviewer feel that interest is limited to structuralists and people interested in the biology of this specific bacterial predator. Despite the high-resolution structures here provided and extensive analysis and comparison with related lytic enzymes, unfortunately the manuscript fails in identification of the molecular basis for recognition of specific NAG-deacetylated PG. Major concerns are detailed below:

1- Line 90. Authors identify two peptidoglycan NAG-deacetylases that, when mutated, allow complete predation but they keep undigested prey cell wall materials. Thus, NAG-deacetylation is not crucial for bacterial release. In agreement, analysis of the *E. coli* bdelloplasts reveals that PG is N-deacetylated (60-70%) in both NAG and NAM saccharide units of PG. While deacetylation seems to be produced in both glycan units, however the present study on DslA is only focused on NAG deacetylation. Could some of the other potential lysozymes in *B. bacteriovorus* present specificity for NAM-deacetylated PG?

2- Line 137. DslA is a lipoprotein, presenting the classical lipobox motif, a flexible region (by the way, what is the length of this region?) and finally the catalytic domain for which the structure is presented. *B. bacteriovorus* being a G(-) bacteria could have DslA lipoprotein attached to the inner or the outer membrane (any information about it?), how authors envision this lipoprotein could act in vivo against the prey cell-wall?

3- Line 150-154. Description of the region for the second catalytic residue would require a better figure. The present figure 3 does not provide information on polar interactions in this region.

4- One of the major flaws in the manuscript is that, contrary to authors claim (line 286-287), the structural analysis does not explain how specific recognition of NAG deacetylation by DslA is achieved. Obviously, a crystallographic complex could immediately answer this question. In its absence, authors have performed a comparison with some previously reported structures. However, this analysis is not properly done; while Figure 3d and 3e shows a comparison of the fold for closely and more distant glycoside hydrolases, analysis of the substrate recognition is performed at the discussion section and then this analysis is not performed in depth:

a. DALI server revealed that closest structural enzymes are lytic transglycosylases (LTs) from family 1 for which a vast structural information has recently being provided (MltE, MltF, Slt, Slt70, MltC...) in complex with some ligands (chitopentaose) and real PG fragments (even with both, glycan and peptide stems). In the present manuscript authors, instead make the comparison with a very old lysozyme structure in which the chitohexaose ligand is hydrolyzed, thus avoiding real indication on the position for the natural substrate in the catalytic crevice.

b. The convention for lysozymes and LTs is that the substrate-binding cleft accommodates several saccharide units at subsites designated as positions -i (the nonreducing end) through +j (in the

other direction). The saccharide units flanking the scissile glycosidic bond are designated as positions -1 and +1. This convention is not followed in the manuscript inducing some errors in the description of substrate recognition by DslA. Thus, the superposition shown in Figure 6c, provides information for the positions -1 to -4, but not for NAG at position +1 or NAM at +2 (as they moved when the substrate was hydrolyzed). Besides, NAM at -1 seems to be a little bit far from the catalytic residue (maybe it also moved a bit to reallocate the substrate at positions +1 and +2).

c. Superposition of DslA with inactive MltE:chitopentaose (PDB code 4HJZ) and inactive Slt in complex with NAG-NAMpentapeptide-NAG-NAMpentapeptide (PDB code 6FCR) or with 4(NAG-NAMpentapeptide) (PDB code 6FCS) could provide a more clear indication on how the different acetyl moieties could affect recognition by DslA.

d. The complex represented in Figure 6c does not indicate how the NAM units can be recognized as this ligand is only composed by NAG units. In this sense, the figure legend "The cell wall sugar rings are labeled a-f, with GlcNAc and MurNAc residues colored blue and green, respectively" is not correct.

e. Comparison with real PG ligands could also provide indication about how recognition of peptide stems could be performed (or not) by DslA in vivo.

5- As indicated in lines 315-319, a single mutant in Tyr228Ala (the only residue mutated in this study, together with catalytic residues) cannot convert DslA in an enzyme competent in turning over acetylated peptidoglycan. Thus, authors suggest that "substrate selection is likely to involve additional factors to steric block by Y228, possibly selecting for the subtly different torsion angles between GlcNAc and GlcN or other differences in recognition." While, I agree, that potential rearrangement of PG cannot be discarded, I think a more in-depth structural study on substrate recognition could provide relevant clues on this matter.

Reviewer #2 (Remarks to the Author):

This manuscript by Harding et al presents the discovery and both preliminary biochemical and structural characterization of a novel lysozyme, DslA. This lysozyme has unique specificity for N-de-acetylated glucosaminyl residues within peptidoglycan. It was discovered in *Bdellovibrio bacteriovorus*, and the recombinant protein was purified from transformed *Escherichia coli*. This lysozyme was suspected to exist given the lifestyle and metabolic activity of *B. bacteriovorus*, but this manuscript presents the first experimental evidence for its existence, and hence underscores its relevance and novelty. The experiments have been conducted with care and attention to appropriate controls. The manuscript has been produced with care and attention to detail; it is concise and easy to read. This reviewer has only a couple of minor suggestions for its improvement.

1. Lines 53-54: Technically, whereas several murolytic enzymes are indeed used for the tailoring of peptidoglycan during growth, division, signaling, autolysis, sporulation, and the insertion of wall-spanning complexes, lysozyme itself is not used for these purposes. Instead, lytic transglycosylases and N-acetylglucosaminidases serve these functions in Gram-negative and Gram-positive bacteria, respectively. Hence, this statements requires some attention.

2. Line 177: The lytic transglycosylases of families GH 102, 103 and 104 also use a single catalytic acid-base residue for catalysis.

Reviewer #3 (Remarks to the Author):

The manuscript proposes a new class of lysozyme-like enzymes in the obligate predator *Bdellovibrio bacteriovorus*. They are characterized by specificity for deacetylated peptidoglycan, and novel structural features. As *B. bacteriovorus* has previously been shown to deacetylate its prey's peptidoglycan upon invasion, the finding of enzymes able to degrade the modified cell wall is not only novel, it also provides a logical answer to the question how progeny cells exit prey. Thus, the manuscript can be viewed as containing two main messages: 1. A novel class of lysozyme-like enzymes were discovered, and they are specific for deacetylated peptidoglycan, and 2. These enzymes are important for in the life cycle of the predatory bacterium. The paper has interesting findings but also in my view, major shortcomings.

Major comments

My major comment is that the biological role of the enzymes is not sufficiently explored – being no expert in structural biology I do not comment on that part of the manuscript. It is made clear a number of times that the aim of the study is restricted to DslA. Although legitimate, it largely reduces the "biological" impact of the manuscript because one of the major aims, as I get it, is to decipher the role of these novel enzymes in the predator's cell cycle. However, by almost exclusively focusing on DslA, this is occulted, and missing.

Without detailing the contribution of each of the genes, the interpretation of the activities of the different enzymes is difficult as DslB and DslC appear to be as active on partially deacetylated peptidoglycan as DslA (Figure 5). This leads to unfounded statements like the DslB and DslC are "in reserve", contradicting "DslB and DslC which were not expressed at exit time" (when in fact, they are, especially dslC, Figure S1). In any case, double and triple mutants of dslA, dslB, dslC are missing to be able to untangle their respective roles in predation at large but also specifically in the premature release from the bdelloplast as tested (figures 7 and supfig 7, 8). Also missing is complementation of delta0314. If these enzymes play a significant role in the life cycle of the predator, then they (or at least DslA) should have an effect on fitness. Therefore, an experiment testing growth rate and or any other fitness parameter should be included.

The parameters used in the microscopic experiment are not well explained. I could not make sense of the different categories, which are not statistically tested. I find it difficult to spot events such as "intermediate", without confirmation by EM. The experiments were set with semi-synchronous cultures. I am certain that the authors know the concentrations of predators and prey used in them and that for setting (semi) synchrony, the ratio of predator to prey is >1 , leaving free predators in suspension. Some may even attach to bdelloplasts. However, there is no mention how this issue is treated to enable counting released progeny.

Expression levels are not quantified and not statistically analysed. While this is barely OK to show temporal shifts during the life cycle, it is not when comparing levels of expression between experiments (i.e. the different genes) (figures 1, S1). One then wonders what Bd1411 is? It is mentioned to be a cryptic lysozyme homologue, it has a differential expression pattern during the life cycle of the predator but nothing additional is written on it. I deduce that it has even lower sequence identity to DslA than DslB (line 214). It is not clear if it was solely identified on the presence of E154 (line 112) or on other sequence attributes as well to include it as a potential lysozyme.

A main finding is the specificity of the Dsl enzymes for deacetylated peptidoglycan. Yet, the experiments were made on a partially deacetylated polymer while a purified fraction should have been used. Furthermore, no statistics are provided in figure 5 and the test itself, as described, appears to be of cumulative activity, not of enzymatic activity (units/time). One would also have expected enzyme parameters to be measured (e.g. K_m , V_{max}).

It is mentioned in the discussion that Dsl-like sequences have been found in other periplasmic predators but not in epibiotic predators; no other details (which strains, how the comparisons were made) are to be found in the manuscript.

Finally, the origin of the novel class of enzymes may have been looked upon. Some sequence alignments are provided (SupFig. 6) but a phylogenetic analysis, including larger classes of enzymes could have been included to shed light on its evolution.

Other concerns are:

No rationale is given for testing HI derivatives and no detail on their growth – are these mixed

cultures of attack phase and growing filaments or something else? Is there a meaning for the difference in levels of expression between HI-22 and the other lines?

In addition to gene mapping (Figure S2b), RT-PCR with primers targeting adjacent genes could be used to show/disprove the presence of a polycystron.

Since the *dsI* genes appear to be all expressed during growth, protection of the bdelloplast from early release of the growing filament, is an interesting issue. Along these lines, one would expect that the addition of exogenous DslA should yield many prematurely released filaments from bdelloplasts, based on the mentioned Ruby et al. experiment.

Response to Reviewers

We thank the reviewers for their time and skill in assessing our work, and provide a point-by-point response (marked with #####, in red) below. We have also highlighted these changes in the manuscript in red.

Reviewer #1 (Remarks to the Author):

The manuscript by Harding et al describes the structural determination of lysozyme DslA from the bacterial predator *Bdellovibrio bacteriovorus*. During release of the bacterial progeny this bacterium uses deacetylases to modify the cell wall (peptidoglycan, PG) from the prey, and subsequently hydrolyze it by the use of specific lysozymes. The present work associates such lysozyme with lipoprotein DslA and presents its three-dimensional structure for the catalytic domain. The manuscript is clear and methodology sound. The structure of the wt and two mutants for the catalytic residues are provided at high-resolution revealing a fold related with that of hen egg-white lysozyme (HEWL) and GH23 family of family 1 of Lytic transglycosylases (LTs). While some bacteria modify its PG by deacetylation in order to avoid recognition by lysozymes from the host, bacterial predators such as bacteriophages or *B. bacteriovorus* have developed specific lysozymes that overcome such problem in order to release their progeny. The manuscript is well written and all experiments seem to have been performed well.

We thank the reviewer for this general appreciation of our work.

However, this reviewer feel that interest is limited to structuralists and people interested in the biology of this specific bacterial predator. Despite the high-resolution structures here provided and extensive analysis and comparison with related lytic enzymes, unfortunately the manuscript fails in identification of the molecular basis for recognition of specific NAG-deacetylated PG. Major concerns are detailed below:

1- Line 90. Authors identify two peptidoglycan NAG-deacetylases that, when mutated, allow complete predation but they keep undigested prey cell wall materials. Thus, NAG-deacetylation is not crucial for bacterial release. In agreement, analysis of the *E. coli* bdelloplasts reveals that PG is N-deacetylated (60-70%) in both NAG and NAM saccharide units of PG. While deacetylation seems to be produced in both glycan units, however the present study on DslA is only focused on NAG deacetylation. Could some of the other potential lysozymes in *B. bacteriovorus* present specificity for NAM-deacetylated PG?

We are unable to model/infer any such specificity from the primary sequences of uncharacterized *Bdellovibrio* lysozyme homologues, but this is beyond the remit of our study – we are looking for lysozymes that enable release (*e.g.* Figure 2 for delayed release when knocked out, Figure 5 for early release when added exogenously to cells), rather than a full characterization of all lysozyme-like genes. We have therefore avoided speculation on NAM metabolism.

2- Line 137. DslA is a lipoprotein, presenting the classical lipobox motif, a flexible region (by the way, what is the length of this region?)

An approximate value for this region is 35 AA; we now include this information on p.7 line 163

and finally the catalytic domain for which the structure is presented. *B. bacteriovorus* being a G(-) bacteria could have DslA lipoprotein attached to the inner or the outer membrane (any information about it?), how authors envision this lipoprotein could act *in vivo* against the prey cell-wall?

We do not have experimentally-determined localization information for DslA, but the need to work on prey wall is highly suggestive of outer membrane tethering; supportive of this, the Ruby et al study we reference finds lytic activity in a predation supernatant wherein the (presumed) DslA component would be present at the outer membrane, as noted on line 285 of the manuscript.

3- Line 150-154. Description of the region for the second catalytic residue would require a better figure. The present figure 3 does not provide information on polar interactions in this region.

We agree and now include a larger figure 3 with additional information on the surrounding residues and the hydrogen bonding patterns.

4- One of the major flaws in the manuscript is that, contrary to authors claim (line 286-287), the structural analysis does not explain how specific recognition of NAG deacetylation by DslA is achieved. Obviously, a crystallographic complex could immediately answer this question. In its absence, authors have performed a comparison with some previously reported structures. However, this analysis is not properly done; while Figure 3d and 3e shows a comparison of the fold for closely and more distant glycoside hydrolases, analysis of the substrate recognition is performed at the discussion section and then this analysis is not performed in depth:

- a. DALI server revealed that closest structural enzymes are lytic transglycosylases (LTs) from family 1 for which a vast structural information has recently being provided (MltE, MltF, Slt, Slt70, MltC...) in complex with some ligands (chitopentaose) and real PG fragments (even with both, glycan and peptide stems). In the present manuscript authors, instead make the comparison with a very old lysozyme structure in which the chitohexaose ligand is hydrolyzed, thus avoiding real indication on the position for the natural substrate in the catalytic crevice.

As noted below, comparisons with Lytic transglycosylases (e.g. PDB codes 4HJZ, 6FCS) result in an identical model for discrimination as our chitohexose model, and we now include these in a new expanded Supplementary Figure 12. Similarity in position of the DslA Y228 OH and modelled NAG acetyl oxygen (in all three examples) lends confidence in our model of steric occlusion.

- b. The convention for lysozymes and LTs is that the substrate-binding cleft accommodates several saccharide units at subsites designated as positions -i (the nonreducing end) through +j (in the other direction). The saccharide units flanking the scissile glycosidic bond are designated as positions -1 and +1. This convention is not followed in the manuscript inducing some errors in the description of substrate recognition by DslA. Thus, the superposition shown in Figure 6c, provides information for the positions -1 to -4, but not for NAG at position +1 or NAM at +2 (as they moved when the substrate was hydrolyzed). Besides, NAM at -1 seems to be a little bit far from the catalytic residue (maybe it also moved a bit to reallocate the substrate at positions +1 and +2).

Our new supplementary Figure 12 reveals that our model for occlusion is identical when either using an intact (PDB 4HJZ) or cleaved (PDB 6FCS) set of co-ordinates for comparison. Sequence conservation of the YW motif is also supportive of this.

- c. Superposition of DslA with inactive MltE:chitopentaose (PDB code 4HJZ) and inactive Slt in complex with NAG-NAMpentapeptide-NAG-NAMpentapeptide (PDB code 6FCR) or with 4(NAG-NAMpentapeptide) (PDB code 6FCS) could provide a more clear indication on how the different acetyl moieties could affect recognition by DslA.

We thank the reviewer for this suggestion and now include these in Supplementary Figure 12; we had used them originally in our analysis prior to manuscript preparation, but chose the singular

1LZS as a well-known/referenced example – we are happy now to include these multiple comparisons, although our conclusions remain the same.

- d. The complex represented in Figure 6c does not indicate how the NAM units can be recognized as this ligand is only composed by NAG units. In this sense, the figure legend “The cell wall sugar rings are labeled a-f, with GlcNAc and MurNAc residues colored blue and green, respectively” is not correct.

They are chemically NAG yes, but spatially equivalent to NAM (this assumption is universal e.g. in the original papers utilizing chitohexose, it is discussed as so – the 1LZS reference states that the chitohexose saccharide positions are largely identical with those of the NAM-NAG-NAM ligand of Strynadka & James; other authors agree also e.g. from the Boneca lab doi: 10.1074/jbc.RA117.001095 “the chitin polymer mimics the chemical composition of the PG glycan strand”).

- e. Comparison with real PG ligands could also provide indication about how recognition of peptide stems could be performed (or not) by DslA in vivo.

Because of the structure divergence from known lysozymes outside of the active site cleft (Figure 3), we feel this might be too speculative for inclusion in the manuscript.

5- As indicated in lines 315-319, a single mutant in Tyr228Ala (the only residue mutated in this study, together with catalytic residues) cannot convert DslA in an enzyme competent in turning over acetylated peptidoglycan. Thus, authors suggest that “substrate selection is likely to involve additional factors to steric block by Y228, possibly selecting for the subtly different torsion angles between GlcNAc and GlcN or other differences in recognition.” While, I agree, that potential rearrangement of PG cannot be discarded, I think a more in-depth structural study on substrate recognition could provide relevant clues on this matter.

We have indeed attempted to obtain substrate complexes (as noted on p.13, line 321), but have been unsuccessful (so far), potentially because of the heterogeneity of chitosan and the weak binding of shorter glucosamine fragments. Future efforts beyond the scope of this current study will look at covalently trapping complexes.

Reviewer #2 (Remarks to the Author):

This manuscript by Harding et al presents the discovery and both preliminary biochemical and structural characterization of a novel lysozyme, DslA. This lysozyme has unique specificity for N-de-acetylated glucosaminyl residues within peptidoglycan. It was discovered in *Bdellovibrio bacteriovorus*, and the recombinant protein was purified from transformed *Escherichia coli*. This lysozyme was suspected to exist given the lifestyle and metabolic activity of *B. bacteriovorus*, but this manuscript presents the first experimental evidence for its existence, and hence underscores its relevance and novelty. The experiments have been conducted with care and attention to appropriate controls. The manuscript has been produced with care and attention to detail; it is concise and easy to read.

We thank the reviewer for their positive appraisal.

This reviewer has only a couple of minor suggestions for its improvement.

1. Lines 53-54: Technically, whereas several murolytic enzymes are indeed used for the tailoring of peptidoglycan during growth, division, signaling, autolysis, sporulation, and the insertion of wall-spanning complexes, lysozyme itself is not used for these purposes. Instead, lytic transglycosylases and N-acetylglucosaminidases serve these functions in

Gram-negative and Gram-positive bacteria, respectively. Hence, this statements requires some attention.

We apologise for this technical omission and have reworded the statement to read “usage of lysozyme/lytic transglycosylase activity” at line 54.

2. Line 177: The lytic transglycosylases of families GH 102, 103 and 104 also use a single catalytic acid-base residue for catalysis.

Again, we apologise for this oversight and have amended this to read “or one residue for the GH23/GH102/GH103/GH104 lytic transglycosylases” at line 198.

Reviewer #3 (Remarks to the Author):

The manuscript proposes a new class of lysozyme-like enzymes in the obligate predator *Bdellovibrio bacteriovorus*. They are characterized by specificity for deacetylated peptidoglycan, and novel structural features. As *B. bacteriovorus* has previously been shown to deacetylate its prey's peptidoglycan upon invasion, the finding of enzymes able to degrade the modified cell wall is not only novel, it also provides a logical answer to the question how progeny cells exit prey. Thus, the manuscript can be viewed as containing two main messages: 1. A novel class of lysozyme-like enzymes were discovered, and they are specific for deacetylated peptidoglycan, and 2. These enzymes are important for in the life cycle of the predatory bacterium. The paper has interesting findings but also in my view, major shortcomings.

Major comments

My major comment is that the biological role of the enzymes is not sufficiently explored – being no expert in structural biology I do not comment on that part of the manuscript. It is made clear a number of times that the aim of the study is restricted to DslA. Although legitimate, it largely reduces the "biological" impact of the manuscript because one of the major aims, as I get it, is to decipher the role of these novel enzymes in the predator's cell cycle. However, by almost exclusively focusing on DslA, this is occulted, and missing. Without detailing the contribution of each of the genes, the interpretation of the activities of the different enzymes is difficult as DslB and DslC appear to be as active on partially deacetylated peptidoglycan as DslA (Figure 5).

We focus on DslA as an exemplar of a novel lysozyme with a novel enzymic target involved in *Bdellovibrio* exit from the bdelloplast and include the detailed structural data for this enzyme. We see this as a major new breakthrough. A complete analysis of all *B. bacteriovorus* lysozymes is a massive undertaking beyond the scope of this work.

However, to address the reviewer's concern we interrogated the biological roles of the other lysozymes by single gene deletion (or active site inactivation) of each candidate, coupled with assays of entry into, and exit from the bdelloplast (pages 6-7, newly expanded Fig. 2 & Supplementary Fig. 3). As suggested in our original manuscript, we could confirm that DslA is predominantly involved in the exit phase, while Bd1411 has the strongest effect on prey-entry. Notably Bd1411 is the most distal member to DslA with no characteristically conserved specific amino acids/motifs for the Dsl “family” and hence likely to recognize the ‘usual’ non-modified acetylated GlcNAc in the entry process (lines 112-114 now describing distal nature, also visible in new Supplementary Figure 11 and its legend). DslB and DslC may have minor roles in prey exit, possibly playing a duplicated role as is seen for other *B. bacteriovorus* PG-active enzymes such as DD-endopeptidases and LD-transpeptidases (references 12 and 14 in manuscript).

This leads to unfounded statements like the DslB and DslC are "in reserve", contradicting "DslB and DslC which were not expressed at exit time" (when in fact, they are, especially dslC, Figure S1).

We apologize for this confusion based on our text. On original submission page 5 lines 114-115 (now lines 115-116) we state that DslB (*bd1413*) and DslC (*bd1440*) transcription is not differentially regulated at specific timepoints of the predatory life cycle of *B. bacteriovorus* (as seen in Supplementary Figure 1, we see a more or less constant expression from 45 min to 240 min, which might be slightly more than in attack phase, 15min and 30 min). We agree that there is a certain level of expression at exit time and are grateful to the reviewer to bring this to our attention. We therefore changed the statement by inserting a "differentially" before "expression" to revise the original statement (current page 15 line 364).

In any case, double and triple mutants of dslA, dslB, dslC are missing to be able to untangle their respective roles in predation at large but also specifically in the premature release from the bdelloplast as tested (figures 7 and supfig 7, 8). Also missing is complementation of delta0314. If these enzymes play a significant role in the life cycle of the predator, then they (or at least DslA) should have an effect on fitness. Therefore, an experiment testing growth rate and or any other fitness parameter should be included.

We have addressed the roles of the other lysozymes as detailed in the answer above by single deletion mutants of DslA and DslC, as well as inactivation of the active site in DslB. Generation of double and triple mutants is a massive undertaking beyond the scope of this study and we were pleased to be able to complete this single gene inactivation study before our labs were closed by the Covid SARS2 lockdown. The extra work confirms that DslA is the major novel lysozyme involved in prey exit.

As the investigation of the biological role of DslA, DslB and DslC at entry and exit are now included in the manuscript (newly expanded Fig. 2 & Supplementary Fig. 3) and *in vitro* enzymatic assays on all of them were performed (Fig. 4) we focused on DslA for the premature release from bdelloplast assay (Fig. 5, Suppl. Figs. 7,8,9 & 10), as DslA plays a major role in exit, while DslB and DslC only play minor roles (Fig. 2). Complementation studies of DslA are now included in the same figure (page 32, expanded Fig. 2, commented in manuscript page 6 lines 140-145), supporting its role in prey exit. Studies on the evolutionary fitness of mutant strain populations would be interesting, but are a large undertaking beyond the scope of this work, which focuses on the mechanism and roles of the enzymes.

The parameters used in the microscopic experiment are not well explained. I could not make sense of the different categories, which are not statistically tested.

An in-depth description of the different categories after bdelloplast treatment with DslA are presented in Supplementary Fig. 8 as mentioned in figure legend of Fig. 5. The different categories (merged as described in Suppl. Table 4) in Fig. 5 (and Suppl. Fig. 9) were statistically tested using Mann-Whitney test and significance levels are indicated within the respective figures and figure legends. However, based on the reviewers comment we noticed, that in Fig. 4 (describing the enzyme assay of DslA) the statistical test was missing, which we now include.

I find it difficult to spot events such as "intermediate", without confirmation by EM.

We agree that "intermediate" events do not necessary represent a prematurely exiting *Bdellovibrio*. Therefore we chose the wording "This finding was accompanied by a concomitant increase in predatory cells, that **appear** to be just in the process of being released (Fig. 5c, *Bdellovibrio* interm.)." (current line 261; page 11). These "intermediate" events only represent a

very small fraction (~5%) and so we concentrated on the 12 - 53 % of all bdelloplasts that were without *Bdellovibrios* (category “*Bdellovibrio* outside”) in 5 independent biological repeats, showing clearly a significant premature release by DslA treatment (Fig. 5).

The experiments were set with semi-synchronous cultures. I am certain that the authors know the concentrations of predators and prey used in them and that for setting (semi) synchrony, the ratio of predator to prey is >1 , leaving free predators in suspension. Some may even attach to bdelloplasts. However, there is no mention how this issue is treated to enable counting released progeny.

The reviewer is correct, that it can not be distinguished if the bdellovibrios imaged by light microscopy were prematurely released from the bdelloplast or originate from the suspension and never invaded prey. As we have no means of telling the fate of individual bdellovibrios, we chose instead to categorise the bdelloplasts (Suppl. Fig. 8, categories c, d, e & f) that had no mCherry-labelled *Bdellovibrios* in them as “*Bdellovibrio* outside”, not by counting released progeny. Based on the generally round peptidoglycan-structure and the presence of the prey cytosol in the bdelloplast, we could confidently assign those with an absence of a mCherry-labelled *Bdellovibrio* as those that have released a previously *Bdellovibrio* inside. To clarify this issue, we now address this at the end of figure legend of Suppl. Fig. 8 (page 10, lines 183-185).

Expression levels are not quantified and not statistically analysed. While this is barely OK to show temporal shifts during the life cycle, it is not when comparing levels of expression between experiments (i.e. the different genes) (figures 1, S1).

Our intention was not to compare levels between genes, only to show temporal shifts. We have changed the phraseology to reflect this (page 31, line 784). Semi-quantitative RT-PCR to show temporal shifts has been validated by quantitative RT-PCR previously (Evans et al., 2007, Lambert et al., 2010). We believe our data strongly supports our conclusions that *bd0314* is expressed later in the predation cycle, *bd1411* is expressed early in predation and that *bd1413* and *bd1440* are expressed throughout the predatory lifecycle (Suppl. Fig. 1) and that these expression patterns match their proposed roles in predation.

One then wonders what Bd1411 is? It is mentioned to be a cryptic lysozyme homologue, it has a differential expression pattern during the life cycle of the predator but nothing additional is written on it. I deduce that it has even lower sequence identity to DslA than DslB (line 214). It is not clear if it was solely identified on the presence of E154 (line 112) or on other sequence attributes as well to include it as a potential lysozyme.

We apologise for any confusion here. Bd1411 is the most distally related of this grouping of enzymes. We have now moved a homology statement (from line 213-215, now placed earlier on p. 5 at start of results section, lines 111-114), adding to it to state “four divergent/cryptic lysozyme homologues are encoded, Bd0314 (DslA), Bd1413 (DslB), Bd1440 (DslC), and a more distal member Bd1411 (22% identity to DslA, in comparison to DslB and DslC which are 30% identical to DslA and 60% identical to each other).” Our newly added analysis ascribes a role for Bd1411 in prey entry (lines 127-132 on page 6 in results, lines 375-376 on page 15 in discussion). We also added a Supplementary Maximum likelihood evolutionary tree (Suppl. Fig. 11) which more clearly shows the relationship between Bd1411 and the other lysozyme homologues.

A main finding is the specificity of the Dsl enzymes for deacetylated peptidoglycan. Yet, the experiments were made on a partially deacetylated polymer while a purified fraction should have been used.

Practical considerations limit us here – we are generating the substrate using *Bdellovibrio* recombinant enzymes, whose activity is limited when taken out of the predatory system they are usually deployed in. Full deacetylation may not be the true endpoint either in the context of the bdelloplast. Unfortunately, purifying substrate for this assay is challenging because of the complexity of the polymer and non-stoichiometric nature of the deacetylation resulting in it being impossible to separate deacetylated cell wall from the intact material (see further discussion of this below). We also note that this is standard for analysis of chitosanases given that chitosan is also a heterogenous polymer. We have made our experimentation as biologically relevant as possible and feel that this is the best approach.

Furthermore, no statistics are provided in figure 5 and the test itself, as described, appears to be of cumulative activity, not of enzymatic activity (units/time). One would also have expected enzyme parameters to be measured (e.g. K_m , V_{max}).

The analysis of peptidoglycan degrading enzymes is limited by the access to chemically defined substrates and unfortunately as a result of this full kinetic parameters (K_m , V_{max}) are not reported for all but a few specialist studies (e.g. Blackburn and Clarke 2002, doi.org/10.1021/bi011833k). Even in those studies, these parameters have limited meaning given the complete insolubility of peptidoglycan and the chemical variability of peptidoglycan between batches and within a single batch. As a result of this, activity assays are of the standard used in many cell wall studies, which are often limited to probing substrate specificity. The FITC-based assay does however improve on existing methodology (turbidimetry or dye-release are both more frequently employed). The FITC reporter we are using is more sensitive than turbidimetry and dye-release; the filtration step we use to remove the substrate leads to lower background and increased reproducibility. Regarding the statistics, we have now reported t-tests in an improved Figure 4.

It is mentioned in the discussion that Dsl-like sequences have been found in other periplasmic predators but not in epibiotic predators; no other details (which strains, how the comparisons were made) are to be found in the manuscript.

We apologise and have added a sentence thus: “A search of the genomes of the model epibiotic predators *B. exovor* and *Micavibrio aeruginosavorus* revealed no DslABC homologues (no hits with an E value < 0.1).” at lines 356-358.

Finally, the origin of the novel class of enzymes may have been looked upon. Some sequence alignments are provided (SupFig. 6) but a phylogenetic analysis, including larger classes of enzymes could have been included to shed light on its evolution.

We thank the reviewer for their comment which improved our manuscript. We implemented at Supplementary Figure 11 a phylogenetic tree on the new Dsl class of lysozymes and included more distantly related groups of lysozymes and related enzymes.

Other concerns are:

No rationale is given for testing HI derivatives and no detail on their growth – are these mixed cultures of attack phase and growing filaments or something else? Is there a meaning for the difference in levels of expression between HI-22 and the other lines?

HI growth mimics intraperiplasmic growth, so it was intriguing to see if this lysozyme was still expressed in these conditions, despite not having a prey cell wall substrate upon which to act. The-growth conditions have been added to the Supplementary Figure 2 legend, page 3. Variability of expression levels are common between different HI strains, also noted and now referenced on Supplementary Figure 2 legend, page 3, lines 52-53).

In addition to gene mapping (Figure S2b), RT-PCR with primers targeting adjacent genes could be used to show/disprove the presence of a polycystron.

This is a minor part of the manuscript from which no strong conclusions were drawn, so further investigation beyond that presented was not completed.

Since the *dsl* genes appear to be all expressed during growth, protection of the bdelloplast from early release of the growing filament, is an interesting issue. Along these lines, one would expect that the addition of exogenous DslA should yield many prematurely released filaments from bdelloplasts, based on the mentioned Ruby et al. Experiment.

In our early bdelloplast-treatment assay (Fig. 5), we used bdelloplasts 1 hour post-mixing of *Bdellovibrio* and prey. At this stage, the internal *Bdellovibrio* have only just begun to grow and so do not appear as long filaments after an additional hour of treatment by mixing with DslA or controls (in total 2 hours post infection). Attempts to start external lysozyme treatment later in the predatory life cycle was not carried out as it would carry the risk that the activity of externally added DslA might be blurred (impeding interpretation) by the later timed natural action of bdelloplast-internal DslA in the exit process. So we agree with the reviewer that if we could work on later stage bdelloplasts then we would see longer filaments like Ruby, but we confined our experimentation to early stage bdelloplasts so we would not be encroaching on natural exit timed protein expression in the bdelloplasts.

References:

EVANS, K. J., LAMBERT, C. & SOCKETT, R. E. 2007. Predation by *Bdellovibrio bacteriovorus* HD100 requires type IV pili. *Journal of Bacteriology*, 189, 4850-9.

LAMBERT, C., CHANG, C. Y., CAPENESS, M. J. & SOCKETT, R. E. 2010. The first bite-profiling the predatosome in the bacterial pathogen *Bdellovibrio*. *PLoS One*, 5, e8599.

Many thanks for your consideration of our revised manuscript,

Dr Andrew Lovering, University of Birmingham

Professor R Liz Sockett FRS, University of Nottingham

REVIEWERS' COMMENTS:

Reviewer #1 (Remarks to the Author):

In this revised version authors tried to respond some of my previous issues. While current version presents a better description of the structural results, I think that my main concern still remains; i.e. how the 3D structure of DslA explains specific recognition by this lysozyme for de-acetylated GlcNAc?

On page 13th authors claim that "our structure allows us to explain how this is achieved using a variant on the lysozyme superfamily fold"

Indeed authors find four unique regions not shared with other lysozyme subfamily (by the way these regions should be clearly highlighted in any of the structural images) but contribution of these regions in specificity is not clear and, at last, the potential specificity of DslA for GlcN over GlcNAc presumably arises via steric blockage by two residues, Tyr228 and Met150, of ring at positions -2 and +1 (respectively) if the sugar groups are acetylated (pg. 14, line 335).

As observed in Supplementary Figure 13, Met150 is not conserved at all in DslA relatives and all kind of different residues (polar, hydrophobic, acidic, bulky, small..) appear in such position. Thus, the only responsible for recognition of GlcN would be the conserved Tyr228 (recognition of a specific modification in a long polymeric substrate performed in a single position of the chain and by a single residue). However substitution of Tyr228 by Ala residue (the residue present at this position in conventional lysozymes) "does not "convert" to an enzyme competent in turning over acetylated peptidoglycan" (pg. 14, line 348). Thus, the role of this residue in recognition of GlcN is not evident, and we have, thus, no clear structural explanation for the main activity of this kind of enzyme.

Explanation could come from some kind of structural rearrangement (maybe thanks to these unique regions in DslA) triggered by the substrate or from other kind of combined effects. However, this information is not present in this work and, in my opinion, the present study does not answer the main question of this manuscript nor supports the authors' claim.

Reviewer #3 (Remarks to the Author):

I praise the authors for the thorough and precise revision of their manuscript. They did a great job. The findings are a great contribution to closing the circle of the unique "peptidoglycan story" started some years ago.

Accordingly, I enthusiastically support publication.

REVIEWERS' COMMENTS:

Reviewer #1 (Remarks to the Author):

In this revised version authors tried to respond some of my previous issues. While current version presents a better description of the structural results, I think that my main concern still remains; i.e. how the 3D structure of DslA explains specific recognition by this lysozyme for de-acetylated GlcNAc? On page 13th authors claim that “our structure allows us to explain how this is achieved using a variant on the lysozyme superfamily fold” Indeed authors find four unique regions not shared with other lysozyme subfamily (by the way these regions should be clearly highlighted in any of the structural images)

**** We thank reviewer 1 for their time and suggestions, and have taken this on board, adding an extra panel to supplementary figure 12 (panel C), and amending the legend to detail this.

but contribution of these regions in specificity is not clear and, at last, the potential specificity of DslA for GlcN over GlcNAc presumably arises via steric blockage by two residues, Tyr228 and Met150, of ring at positions -2 and +1 (respectively) if the sugar groups are acetylated (pg. 14, line 335).

As observed in Supplementary Figure 13, Met150 is not conserved at all in DslA relatives and all kind of different residues (polar, hydrophobic, acidic, bulky, small..) appear in such position.

We identify Y228 via analysis of residue conservation at the active site cleft, in the context of comparing DslA close homologues to lysozymes of “normal” specificity (i.e. looking for DslA subfamily-specific features). Gratifyingly, this tallies with knowledge of several lysozyme:substrate complexes in that acetate groups face inward, at distinct locations, and this overlaps with the Y228 sidechain, and at another site, M150 also.

There is an excellent reason for the relative importance of conserving Y228 but having M150 more variable – because Y228 is at position -2 and M150 at position +1, it follows that one can use the model and ask which of this positions has the potential to afford more discrimination. Using the 4HJZ lysozyme substrate complex as an example, analysis with PISA reveals the -2 NAG sugar ring buries 241 Å² surface area, whereas the +1 sugar only buries 185 Å². Hence (intuitively), the greatest contact is made to the middle ring rather than the ends, and this is where DslA divergence is most important. To communicate this idea to the readership we now include the statement “The greater degree of conservation at Y228 rather than M150 is in agreement with analysis of the buried-surface area of lysozyme:substrate complexes wherein more contacts are made to ring c (-2 position) rather than ring e (+1). Hence, DslA is also likely to follow this pattern, discriminating most strongly via the central region of the active site cleft” at line 332.

We agree that absolute conservation of Met150 is not obligatory for acetate-specificity, but the lysozyme subsite model we present is suggestive that the presentation of a sidechain at this position, and the local environment around it, is important.

Our original text on lines 336-339 noted this: “The positioning and absolute conservation of Y228 appears to be more important than the more variable M150 (Supplementary Fig. 13), and is of interest when comparing to the wider lysozyme superfamily” but we have made this more apparent by altering to “The positioning and absolute conservation of Y228 appears to be more important than the more variable M150 (Supplementary Fig. 13, and is of interest when comparing to the

wider lysozyme superfamily. Residues in the vicinity of M150, including the partly conserved Y152, may assist in collectively creating an environment that favors deacetylated substrate.”

Thus, the only responsible for recognition of GlcN would be the conserved Tyr228 (recognition of a specific modification in a long polymeric substrate performed in a single position of the chain and by a single residue). However substitution of Tyr228 by Ala residue (the residue present at this position in conventional lysozymes) “does not “convert” to an enzyme competent in turning over acetylated peptidoglycan” (pg. 14, line 348). Thus, the role of this residue in recognition of GlcN is not evident, and we have, thus, no clear structural explanation for the main activity of this kind of enzyme.

**** Importantly, Y228 is in a region of DslA that has a larger and different topology loop than other lysozyme superfamily members and the backbone carbonyl projects out into the active site cleft, hence mutation to alanine will not remove all elements of the steric block. For clarity, we now add this in on line 351 thus: “One of these factors may be the topology of the peptide backbone in this region; the carbonyl of Y228 projects out relatively further than equivalent residues in the superfamily, and mutation to alanine will not nullify this feature.”.

Explanation could come from some kind of structural rearrangement (maybe thanks to these unique regions in DslA) triggered by the substrate or from other kind of combined effects. However, this information is not present in this work and, in my opinion, the present study does not answer the main question of this manuscript nor supports the authors’ claim.

**** The lack of specificity “reversal” in the Y228A mutant is not surprising given the backbone differences outlined above. Further to this, evolution has optimized the DslA fold for its different specificity in many ways that are less straightforward to investigate in full e.g. pocket hydration, dynamics, shape, charge etc. Quite often in enzymology there is no “simple” answer and we do our best to use language that by no means suggests a definitive singular (sequence-based) reason.

This idea was explained in our original text by the statement on lines 349-351 reading “ thus substrate selection is likely to involve additional factors to steric block by Y228, possibly selecting for the subtly different torsion angles between GlcNAc and GlcN or other differences in recognition”. We have added additional text to stress this complexity so it reads thus: “ thus substrate selection is likely to involve additional factors to steric block by Y228, possibly selecting for the subtly different torsion angles between GlcNAc and GlcN or other differences in recognition. Factors additional to Y228 that could potentially contribute to specificity may be complex and beyond the scope of this study (e.g. active site dynamics, or enzyme/substrate hydration differences in relation to other superfamily members)”.

Reviewer #3 (Remarks to the Author):

I praise the authors for the thorough and precise revision of their manuscript. They did a great job. The findings are a great contribution to closing the circle of the unique "peptidoglycan story" started some years ago. Accordingly, I enthusiastically support publication.

**** We thank reviewer 3 for their time and this strong endorsement of our work.